# Limited dishevelled/Axin oligomerization determines efficiency of Wnt/β-catenin signal transduction

Wei Kan[1,2†], Michael D Enos[1,2†], Elgin Korkmazhan[3,4†], Stefan Muennich[1,2], Dong-Hua Chen[1], Melissa V Gammons[5], Mansi Vasishtha[1,2], Mariann Bienz[5], Alexander R Dunn[3,4], Georgios Skiniotis[1,2], William I Weis[1,2,4*]

[1]Department of Structural Biology and Stanford University School of Medicine, Stanford, United States; [2]Department of Molecular and Cellular Physiology, Stanford University School of Medicine, Stanford, United States; [3]Department of Chemical Engineering, Stanford University, Stanford, United States; [4]Graduate Program in Biophysics, Stanford University, Stanford, United States; [5]MRC Laboratory of Molecular Biology, Cambridge Biomedical Campus, Cambridge, United Kingdom

**Abstract** In Wnt/β-catenin signaling, the transcriptional coactivator β-catenin is regulated by its phosphorylation in a complex that includes the scaffold protein Axin and associated kinases. Wnt binding to its coreceptors activates the cytosolic effector Dishevelled (Dvl), leading to the recruitment of Axin and the inhibition of β-catenin phosphorylation. This process requires interaction of homologous DIX domains present in Dvl and Axin, but is mechanistically undefined. We show that Dvl DIX forms antiparallel, double-stranded oligomers in vitro, and that Dvl in cells forms oligomers typically <10 molecules at endogenous expression levels. Axin DIX (DAX) forms small single-stranded oligomers, but its self-association is stronger than that of DIX. DAX caps the ends of DIX oligomers, such that a DIX oligomer has at most four DAX binding sites. The relative affinities and stoichiometry of the DIX-DAX interaction provide a mechanism for efficient inhibition of β-catenin phosphorylation upon Axin recruitment to the Wnt receptor complex.

*For correspondence:
weis@stanford.edu

†These authors contributed equally to this work

## Introduction

The Wnt/β-catenin signaling pathway mediates cell fate specification during embryogenesis and tissue renewal in the adult (*Nusse and Clevers, 2017*). In this pathway, secreted Wnt proteins activate growth control genes by stabilizing the transcriptional co-activator β-catenin. In the absence of a Wnt signal, a cytosolic pool of β-catenin is bound in a 'destruction complex' that contains the scaffold protein Axin, the kinases GSK-3 and CK1, and the adenomatous polyposis coli protein (APC); phosphorylation of β-catenin leads to its ubiquitylation and destruction by the proteasome (*Stamos and Weis, 2013*). Wnt binding to its cell surface receptors inhibits β-catenin destruction, and the stabilized β-catenin translocates to the nucleus and activates Wnt target genes through association with TCF/LEF transcription factors (*Cadigan and Waterman, 2012*). Inappropriate activation of the pathway by mutations that inactivate β-catenin destruction is associated with a number of cancers (*Clevers and Nusse, 2012*; *MacDonald et al., 2009*; *Nusse and Clevers, 2017*).

Wnt binding to two co-receptors, the 7-transmembrane helix receptor Frizzled (Fzd) and the single pass transmembrane receptor LDL receptor-related protein 5 or 6 (LRP5/6) (*MacDonald and He, 2012*), brings the receptors into close proximity. A current model for signal transduction is that ligand-bound Fzd recruits the cytosolic protein Dishevelled (Dvl), which then binds to Axin and thereby recruits the destruction complex to the activated receptor complex. In this complex, Axin-

**eLife digest** Stem cells can give rise to many types of specialized cells through a process called differentiation, which is partly regulated by changes in the levels of a protein known as β-catenin. On one hand, a 'destruction complex' can keep β-catenin levels low; this complex includes a protein called Axin and an enzyme known as GSK-3, which can tag β-catenin for degradation. On the other hand, when β-catenin levels need to increase, another protein called Dishevelled is activated. By binding to Axin, Dishevelled can bring the destruction complex in contact with other proteins, which leads to the deactivation of GSK-3.

Dishevelled and Axin interact via a region that is similar in the two proteins, called DIX in Dishevelled and DAX in Axin. Studies of DIX and DAX have shown that both regions can form polymers – that is, a high number of similar units can bind together to form larger structures. However, these experiments were at higher concentrations than would be found in the cell. It was thought that, when combined, DIX and DAX might form these long chains together, preventing Axin from carrying out its role in destroying β-catenin. Kan et al. set out to better understand this process by studying how DIX and DAX behave separately, and how they interact.

The proteins were examined using a technique called cryo-electron microscopy, which allows scientists to dissect the structure of large proteins. When there was a high concentration of DIX in the sample, the molecules attached to one another to form long double-stranded helices. Similarly, DAX also formed helices, but these were shorter and only single-stranded. When the two proteins were combined, DAX bound only to the ends of short DIX chains, so that there are not more than four DAX chains attached to each DIX double helix.

To see if this behaviour happens naturally, Kan et al. attached fluorescent tags to Dishevelled proteins and followed them in living cells: this showed that Dishevelled forms smaller chains with fewer than ten molecules. Together these results highlight how Dishevelled binds to Axin to deactivate GSK-3, to prevent the enzyme from promoting β-catenin destruction.

Mutations in the genes that encode β-catenin or its regulators are associated with cancer. Ultimately, a better understanding of how β-catenin is regulated could help to identify new opportunities for drug development.

bound GSK-3 phosphorylates proline/serine-rich motifs in the LRP5/6 cytoplasmic tail, which then inhibit GSK-3 and hence β-catenin phosphorylation and destruction (*MacDonald and He, 2012*; *Stamos et al., 2014*; *Stamos and Weis, 2013*). Dvl self-association may contribute to formation of a multimeric and possibly phase-separated 'signalosome' containing multiple copies of the receptor complex and its associated cytoplasmic components (*Gammons and Bienz, 2018*; *Schaefer and Peifer, 2019*).

The Dvl-Axin interaction is essential for Wnt/β-catenin signal transduction and is mediated by a DIX (Dvl and Axin interacting) domain present in each protein (*Cliffe et al., 2003*; *Fiedler et al., 2011*; *Julius et al., 2000*; *Kishida et al., 1999*; *Smalley et al., 1999*). Dvl has an N-terminal DIX domain, as well as a PDZ and a DEP domain; the DEP domain mediates the interaction with Fzd (*Gammons et al., 2016b*; *Tauriello et al., 2012*). Axin contains binding sites for APC, GSK-3, β-catenin and CK1 (*Behrens, 1998*; *Hart et al., 1998*; *Ikeda et al., 1998*; *Kishida et al., 1999*) and a C-terminal DIX domain. In this paper, we call the Axin DIX domain DAX in order to distinguish it from Dvl DIX (*Bienz, 2014*). In crystal structures, DIX domains from Axin and Dvl form head-to-tail helical polymers (*Liu et al., 2011*; *Madrzak et al., 2015*; *Schwarz-Romond et al., 2007a*; *Yamanishi et al., 2019b*), and Dvl DIX has been seen to form filaments in negative stain electron microscopy (EM) (*Schwarz-Romond et al., 2007a*). Mutations that disrupt the head-to-tail interactions in either protein interfere with signaling, indicating that the DIX-DAX interaction is essential (*Fiedler et al., 2011*). Overexpression of fluorescently tagged Axin or Dvl produces large puncta visible by optical microscopy, and these puncta are lost when these mutations are introduced (*Fiedler et al., 2011*; *Schwarz-Romond et al., 2007a*; *Schwarz-Romond et al., 2005*; *Schwarz-Romond et al., 2007b*). Based on these findings, it has been proposed that Axin self-associates via DAX in the destruction complex. Upon Wnt signaling, it is thought that Dvl self-associates via its DIX domain as well as dimerization of its DEP domain (*Gammons and Bienz, 2018*; *Gammons et al.,*

*2016a*), and subsequently recruits Axin by co-polymerization of the DIX and DAX domains into a filamentous oligomer (*Bienz, 2014*; *Cliffe et al., 2003*). It has also been suggested that the interaction with Dvl may compete with APC for Axin, thereby disrupting the destruction complex (*Fiedler et al., 2011*; *Mendoza-Topaz et al., 2011*).

The extent of filament/oligomer formation by endogenous Dvl and Axin, and how their relative stoichiometries in signaling complexes control GSK-3 activity and β-catenin destruction, are essential but unresolved questions (*Gammons et al., 2016b*; *Schaefer et al., 2018*). Here, we examine the interaction of Axin and Dvl DIX domains, using structural and biochemical analyses of purified DIX and DAX domains, and assessing the role of these domains when the proteins are present at endogenous levels. We found that purified Dvl DIX in solution forms antiparallel, double-stranded filaments whose stability depends on the inter-strand interactions in the double helix. At endogenous expression levels in HEK293T cells, Dvl forms oligomers on the order of 10 molecules. Unlike Dvl, Axin DAX does not form a double-stranded structure, and forms only modestly sized oligomers even though its self-association is approximately 10x stronger than that of DIX. Our evidence indicates that DAX caps the ends of DIX oligomers, such that a DIX oligomer has at most four DAX binding sites. These data suggest molecular mechanisms for efficient inhibition of GSK-3 mediated β-catenin phosphorylation upon Axin recruitment to the Wnt receptor complex.

## Results

### DIX forms an antiparallel double helix that stabilizes filaments

We produced DIX filaments by cleaving purified maltose-binding protein (MBP)-DIX fusion protein, which does not polymerize, with tobacco etch virus (TEV) protease (*Figure 1a*). The purified filaments run as a broad peak in size exclusion chromatography (SEC) with a size range of 4.9–9.5 MDa determined by Multi-Angle Light Scattering (MALS) (*Figure 1b*). The concentration dependence of filament formation was measured by a centrifugation assay in which the relative amount of soluble and pelleted filaments is determined by the intensity of the DIX band on an SDS-PAGE gel (*Figure 1c,d*). Sedimentable filaments were observed starting at a concentration of ~12 µM (*Figure 1c,d*), which is somewhat surprising given estimates of an intrinsic DIX–DIX $K_D$ of 5–20 µM obtained by analytical ultracentrifugation (*Schwarz-Romond et al., 2007a*) and fluorescence polarization of mutants designed to probe dimer formation (*Yamanishi et al., 2019a*).

We obtained a cryo-electron microscopy (cryo-EM) map of the DIX filaments with a global resolution of 3.6 Å, which enabled a near atomic resolution structure (*Figure 2*, *Figure 2—figure supplement 1*, *Table 1*). Remarkably, unlike the single stranded fibers observed in crystals of Dvl2 DIX domains (*Madrzak et al., 2015*; *Yamanishi et al., 2019b*), in solution the Dvl2 DIX filament forms an antiparallel double helix (*Figure 2b,c*). Successive protomers in the filament are related by a rotation of 48.0 ± 0.2° around the helical axis, with a rise of 13.5 ± 0.3 Å per protomer. The pitch of the helix is 101.3 Å, larger than the pitch of the single stranded helix observed in crystals of Dvl2 Y27D (85 Å) and Y27W/C80S (88 Å) (*Madrzak et al., 2015*; *Yamanishi et al., 2019b*). Interestingly, antiparallel packing of Dvl1 Y17D DIX domains were observed in a crystal structure (*Liu et al., 2011*), although the pitch was considerably larger (140 Å).

We verified the accuracy of the structure by mutating residues that mediate contacts between protomers in a single strand, and between strands in the double helix. The head-to-tail interface within each strand is similar to that observed in crystal structures. For example, Y27 packs against F56 and K68 of a neighboring protomer (*Figure 2d*). The mutation Y27D is known to disrupt filament formation (*Liu et al., 2011*), and the purified Y27D mutant runs predominantly as a monomer on SEC (*Figure 3a* and see below). We next tested the role of the inter-strand contacts that form the antiparallel double helix (*Figure 3b*). These contacts are formed between two opposing protomers, principally by the β3-β4 (residues 60–66) and β4-β5 (residues 82–84) loops. Some residues on the β3-β4 loop (e.g., D63) participate in intra- and inter-strand interactions, whereas others appear to be involved solely in inter-strand contacts, including M60, G65 and N82. Mutating each of these last three residues individually reduced the apparent size of the DIX oligomer on SEC (*Figure 3a*), but they run larger than the monomeric Y27D, consistent with their ability to form intra-strand head-to-tail interactions. The G65D mutant, however, runs smaller than the others (*Figure 3a*, *Figure 3—figure supplement 1*), suggesting that the unique conformation of the loop at this position may not

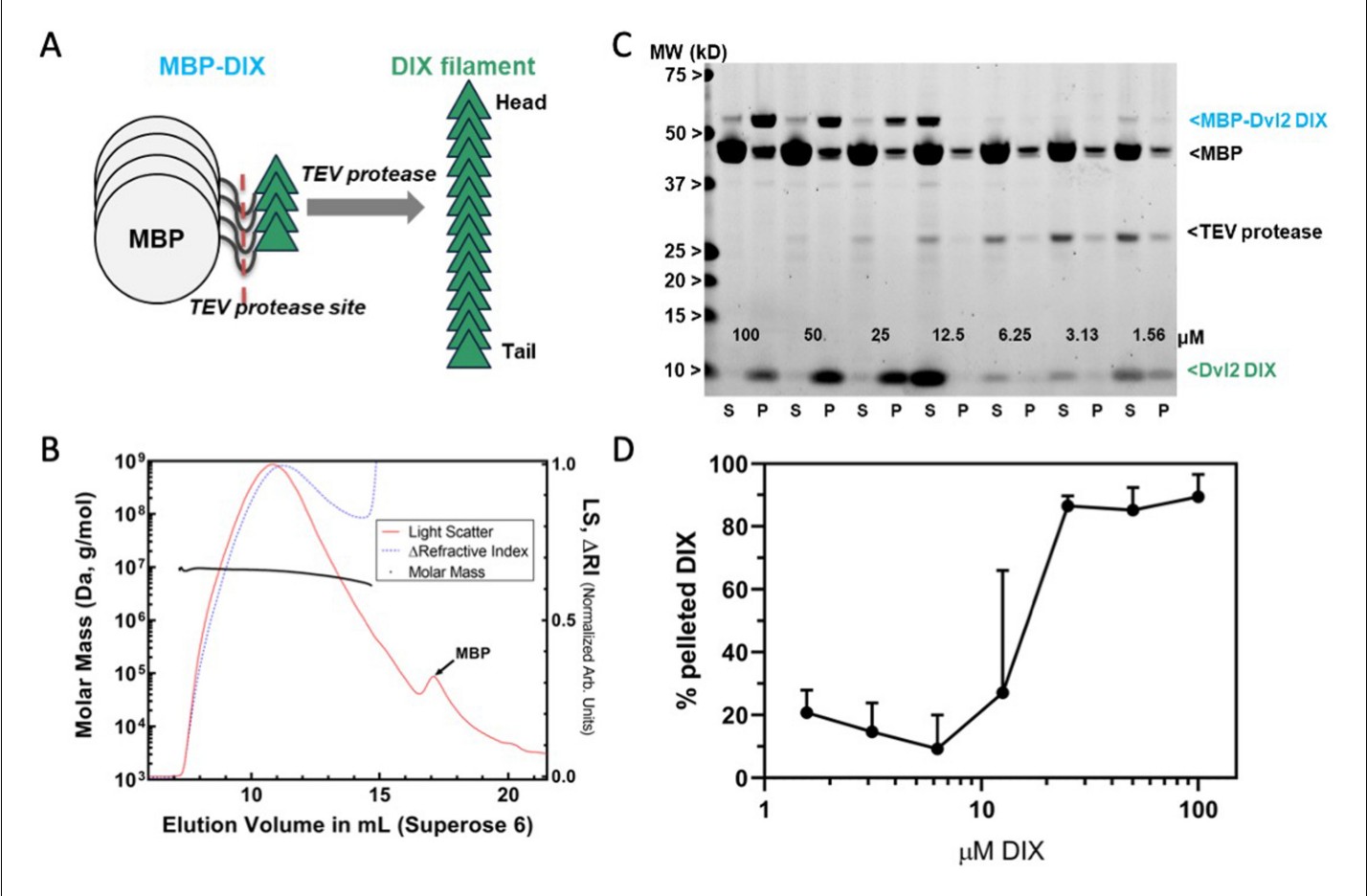

**Figure 1.** Preparation and characterization of Dvl2 DIX filaments. (**A**) Dvl2 DIX filaments were produced by cleavage of the MBP tag with TEV protease. The fusion protein is 52 kDa (MBP is 42 kDa, and the Dvl2 DIX domain is 10 kDa). (**B**) Molecular mass of Dvl2 DIX filaments measured by Multi-Angle Light Scattering coupled to size-exclusion chromatography (SEC-MALS). Dvl2 DIX at ~200 µM produced as in (**A**) was run on a Superose 6 10/300 column in line with MALS and refractive index (RI) detectors. The molar mass (left Y-axis) of Dvl2 DIX filaments was calculated over a range of elution volumes (X-axis), with the average mass ranging from 9.5 MDa for the earliest-eluting species to 4.9 MDa for the latest-eluting species. Residual MBP elutes after DIX filaments. The LS:dRI ratio for DIX filaments is large relative to that of MBP, so the LS and dRI signals have been normalized to show the near congruity of LS and dRI traces for DIX filaments. (**C**) Concentration dependence of Dvl2 DIX filament formation measured by sedimentation. A representative experiment is shown. MBP-Dvl2 DIX at the indicated concentrations was digested with TEV protease, centrifuged, and the supernatant (S) and pellet (P) fractions run on SDS-PAGE. The five highest concentration samples were diluted prior to loading to prevent overloading; since the TEV concentration is constant in each sample, this dilution makes the TEV band intensity different in these lanes. (**D**) Data from six replicates of the experiment shown in (**C**) were quantified and plotted as shown.

tolerate substitution with another residue, or that the mutation might affect head-to-tail interactions as well. Several of these inter-strand mutants were reported earlier, based on antiparallel strand-strand interactions in Dvl1 Y17D crystals (*Liu et al., 2011*) that are very similar to those observed in the present filament structure.

We tested the effects of the Dvl2 DIX structure-based mutants in a luciferase-based reporter (TOPFLASH) assay that measures Wnt-stimulated β-catenin/TCF–stimulated transcription (*Molenaar et al., 1996*), expressing mutant Dvl2 constructs in Dvl triple knockout (TKO) cells (*Gammons et al., 2016b*) cells at near-endogenous levels and normalizing the luciferase signal by the expression level of the mutant relative to wild-type Dvl2 (*Figure 3c*, *Figure 3—figure supplement 2*). As reported previously (*Schwarz-Romond et al., 2007a*), the Y27D mutation reduces signaling to near background levels, even though it can bind DAX (*Yamanishi et al., 2019a*). In contrast, the inter-strand contact mutants reduced signaling substantially, but varied in their level of

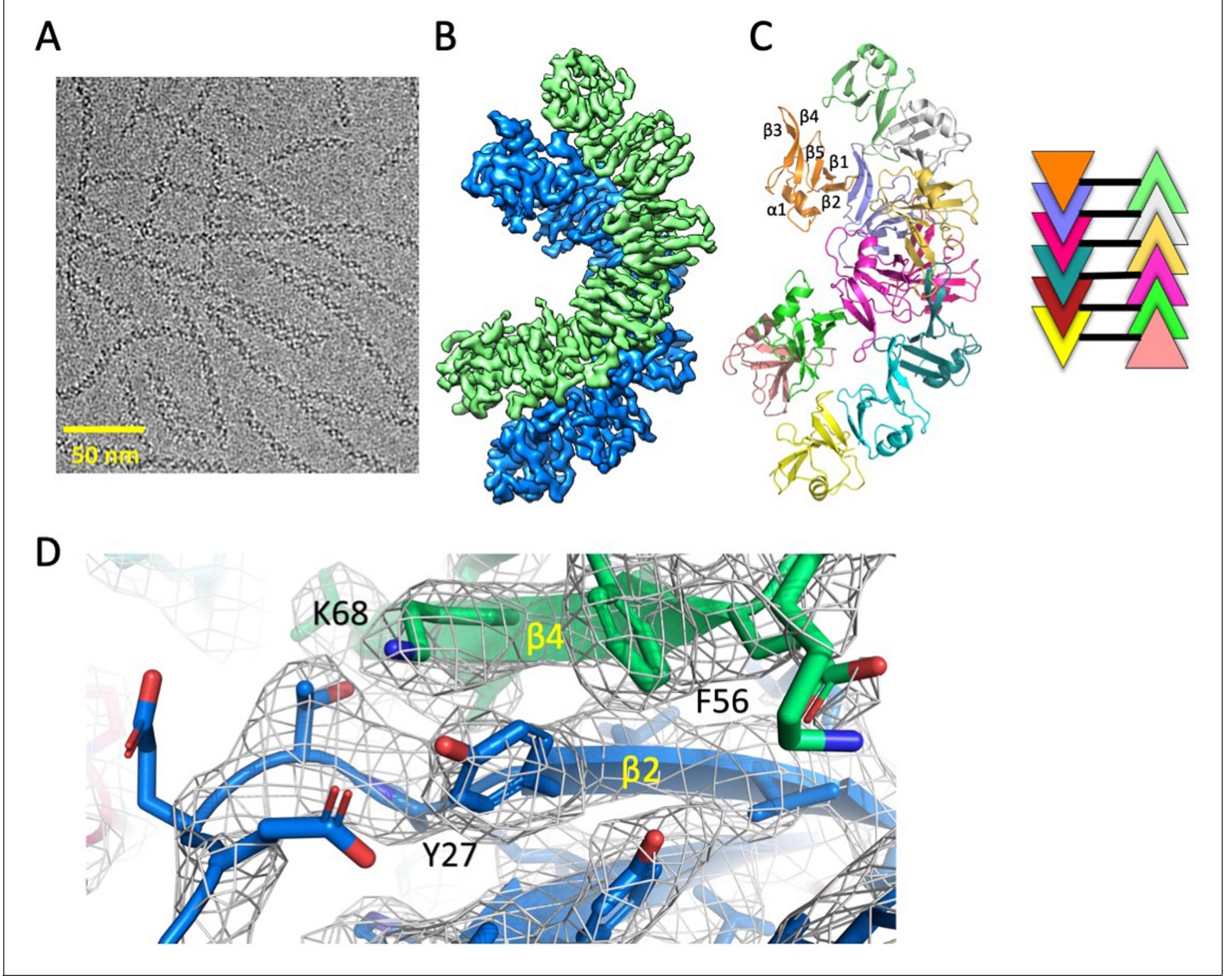

**Figure 2.** Cryo-EM structure of Dvl2 DIX filaments reveals anti-parallel double helices stabilized by intra- strand (head-tail) and inter-strand contacts. (**A**) Cryo-EM image of purified Dvl2 DIX filaments. (**B**) Final sharpened cryo-EM map with helical symmetry imposed, with the two antiparallel strands shown in green and blue contours. (**C**) Final model of the Dvl2 DIX anti-parallel double helix, containing 12 subunits. The secondary structure elements are marked in the upper brown copy. The schematic diagram on the right shows the antiparallel structure, with each subunit represented as a triangle colored as in (**B**). (**D**) A portion of the final cryo-EM map (wire frame) and model showing a head-to-tail interface involving Y27 (head) packing against F56 and K68 (tail).

The online version of this article includes the following figure supplement(s) for figure 2:

**Figure supplement 1.** Cryo-EM analysis of Dvl2 DIX filaments.

suppression (*Figure 3c*), consistent with earlier studies that used overexpression of Dvl2 constructs (*Liu et al., 2011*).

The structural, biochemical and signaling data indicate that the double stranded DIX filament is stabilized by inter-strand, antiparallel contacts that are functionally important. These contacts reinforce the head-to-tail interactions such that large assemblies begin to form at lower concentrations than would be predicted from the head-to-tail $K_D$ alone (*Figure 1c,d*). The data also indicate that a stable oligomer containing these anti-parallel strands is needed for signaling.

**Table 1.** Cryo-EM data collection, refinement and validation statistics.

| Data collection and processing | |
| --- | --- |
| Magnification | 29,000 |
| Voltage (kV) | 300 |
| Electron exposure (e⁻/Å²) | 98 |
| Defocus range (μm) | 0.5–2.0 |
| Pixel size (Å) | 1.00 |
| Symmetry imposed | C1 |
| No. Initial particle images | 437,872 |
| No. Final particle images | 110,105 |
| Map resolution (Å) | 3.6 |
| FSC threshold | 0.143 |
| Map resolution range (Å) | 3.5–4.3 |
| Map sharpening $B$ factor (Å²) | −126 |
| Refinement | |
| Initial model used (PDB code) | 6IW3 |
| Model resolution (Å) | 3.5 |
| FSC threshold | 0.5 |
| Model composition | |
| Chains | 12 |
| Non-hydrogen atoms | 7572 |
| Protein residues | 936 |
| $B$ factors (Å²) | 59.4 |
| R.m.s. deviations | |
| Bond lengths (Å) | 0.004 |
| Bond angles (°) | 0.65 |
| Validation | |
| MolProbity score | 1.47 |
| Clashscore | 4.58 |
| Poor rotamers (%) | 0.00 |
| Ramachandran plot | |
| Favored (%) | 96.4 |
| Allowed (%) | 3.6 |
| Disallowed (%) | 0 |

## Axin DAX oligomerizes through a head-to-tail interface but does not form large polymers

Unlike Dvl2 DIX, Axin DAX runs on SEC in a peak with an abrupt rise and a long tail, at a volume that corresponds to a size much smaller than DIX filaments (*Figure 4a*, *Figure 3—figure supplement 1a*). SEC-MALS analysis revealed that purified DAX forms small oligomers in a concentration-dependent manner. At the highest concentration measured, 24 μM (a concentration at which DIX forms filaments; *Figure 1c,d*), it formed on average octamers (*Figure 4b*, *Table 2*). At the lowest concentration point of 0.7 μM, the average mass of 15 kDa likely represents a mixture of roughly 40% monomer with multiple higher-order oligomers, corresponding to an apparent $K_D$ of about 0.9 μM for the homomeric DAX-DAX interaction (*Table 3*). This is considerably stronger than previously published estimates that employed DAX mutants designed to produce isolated dimers (*Fiedler et al., 2011*; *Yamanishi et al., 2019a*), and importantly, is stronger than the estimated

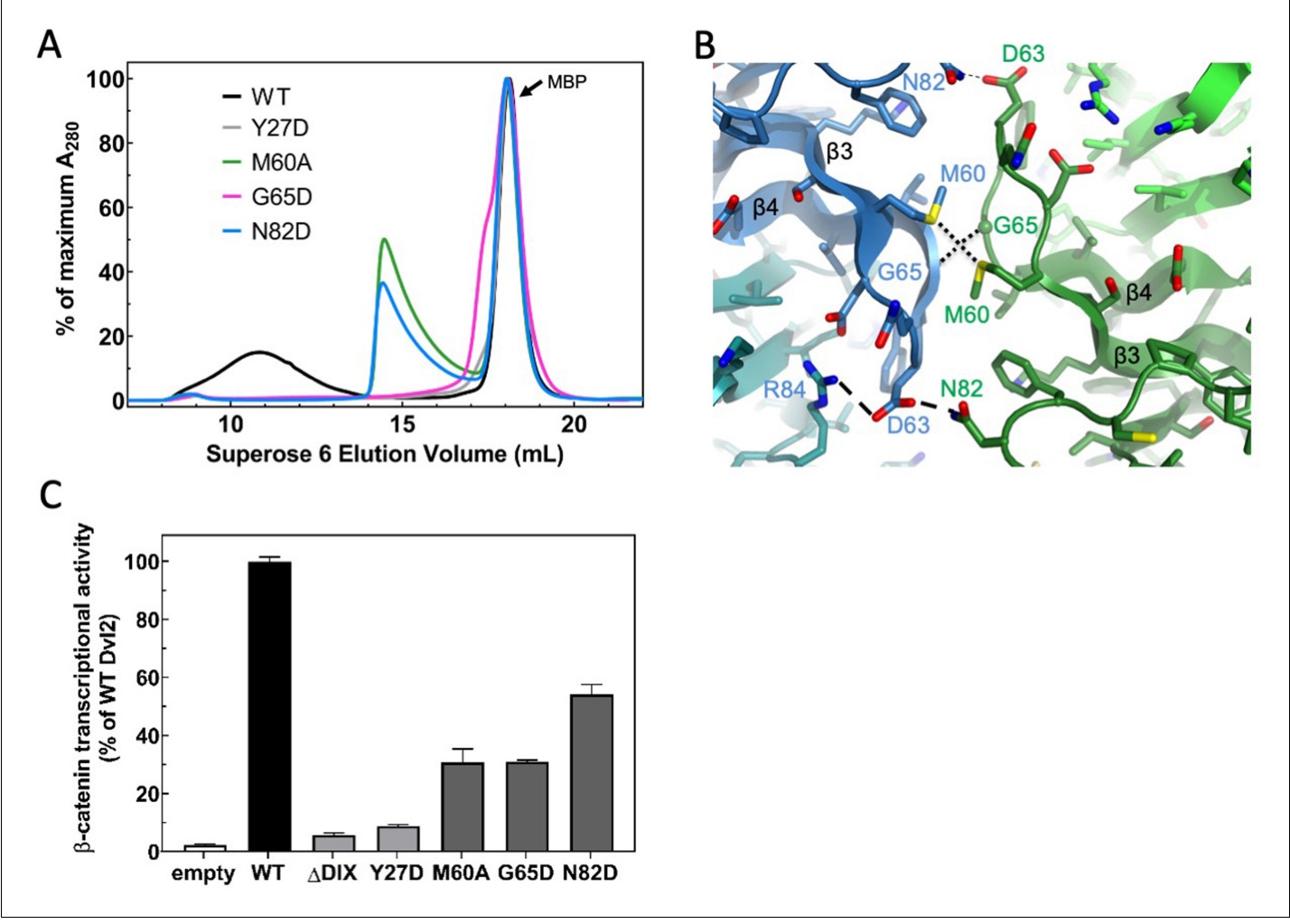

**Figure 3.** Point mutations disrupt double stranded filament formation and Dvl signaling. (A) Comparison of TEV-cleaved Dvl2 DIX variants run on a Superose 6 size-exclusion column. Proteins were injected at 180 µM, and diluted to approximately 15 µM on the column. WT Dvl2 DIX elutes early in a broad peak, whereas point mutants that interfere with filament contacts exhibit varying degrees of oligomerization, manifesting as later elution volumes (smaller sizes). The MBP tag (42 kDa; elution volume = 18 mL) represents the largest peak by $A_{280}$. The separation of the Y27D and G65D from the MBP can be seen on a Superdex 75 column, shown in *Figure 3—figure supplement 1*. (B) Inter-strand contacts within Dvl2 double helix. (C) Mutations of individual residues at intra- (Y27D) and inter- (M60A, G65D, N82D) strand interfaces cause defects in Wnt-dependent β-catenin signaling. Dvl2 constructs are expressed at near endogenous levels in Dvl Triple Knockout (TKO) cell line (see *Figure 3—figure supplement 2* for more details). A construct with the Dvl2 DIX domain deleted (ΔDIX) is also shown.

The online version of this article includes the following figure supplement(s) for figure 3:

**Figure supplement 1.** Purification of WT DAX and size-exclusion chromatography (SEC) of DIX oligomerization mutants.

**Figure supplement 2.** Expression levels of Dvl2 mutants in HEK293T Dvl TKO cells.

homomeric DIX–DIX affinity of 5–20 µM (*Figure 1c,d* and *Schwarz-Romond et al., 2007a*; *Yamanishi et al., 2019a*).

Previous results have suggested that the regions of Axin N-terminally adjacent to its DAX domain, which were named the 'D' (dimerization) and 'I' (inhibition) domains, might modulate Axin oligomerization (*Luo et al., 2005*). SEC-MALS analysis of a purified construct containing the D, I and DAX domains (DI-DAX) indicates that it oligomerizes comparably to WT DAX (*Table 2*). Moreover, SEC-MALS analysis of purified DI-DAX Y760D, which is the mutation equivalent to DIX Y27D, shows it to be monomeric (*Table 2*). Thus, it appears that the D and I regions do not affect the oligomeric state of DAX, and the protomers interact through the head-to-tail interface.

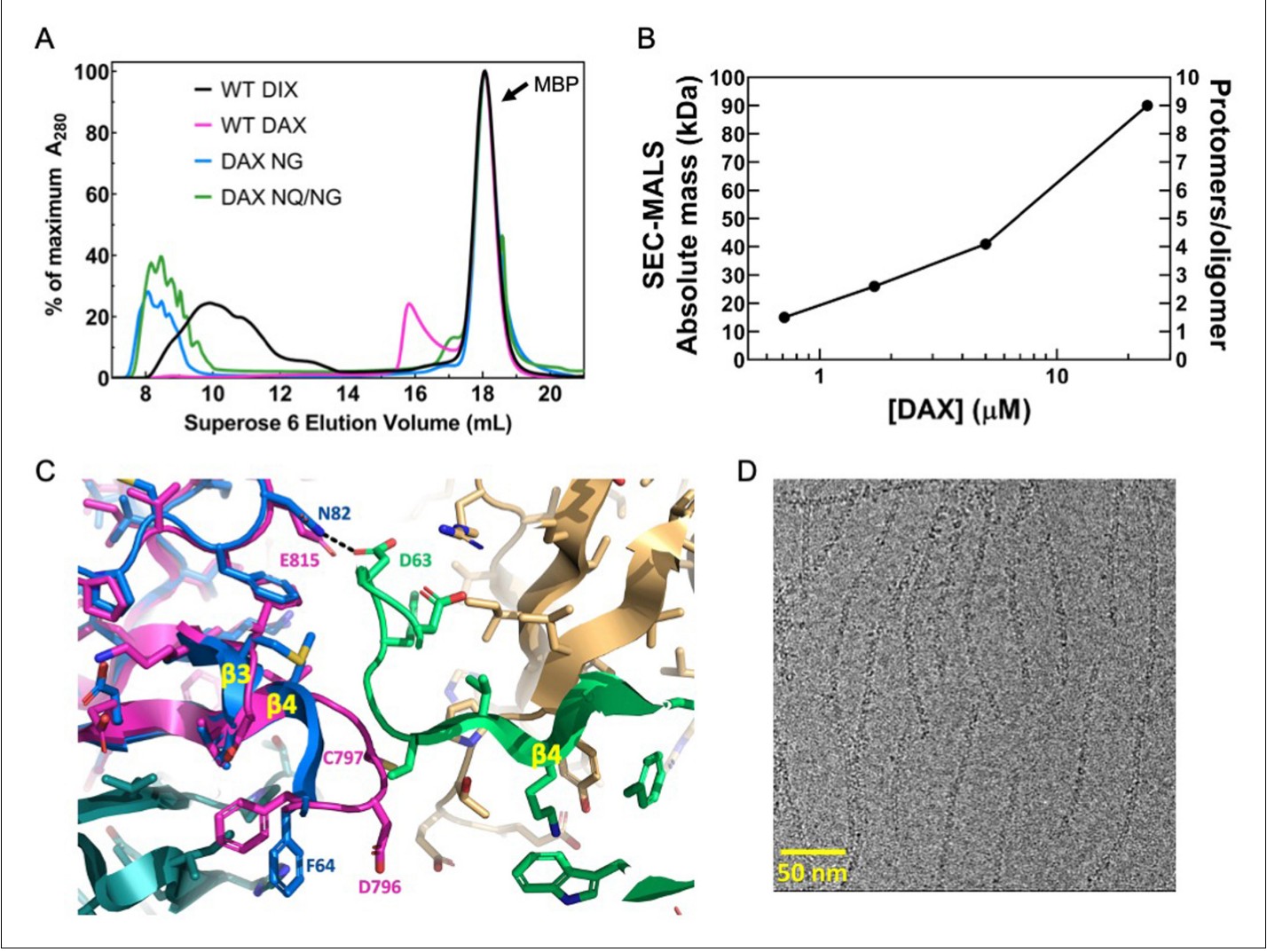

**Figure 4.** Oligomerization of Axin1 DAX and effects of oligomerization on signaling. (**A**) Comparison of TEV-cleaved Axin1 DAX and Dvl2 DIX variants run on a Superose six size exclusion column (NG = E815N/E816G, NQ/NG = D793N/E794Q/E815N/E816G). Proteins were injected at 180 µM, and diluted to approximately 15 µM on the column. (**B**) SEC-MALS analysis of Axin1 DAX size as a function of concentration. (**C**) Superposition of the rat Axin1 crystal structure (PDB 1WSP; magenta) on one of the Dvl2 DIX domains (blue), showing potential clashes with another DIX protomer (green) across the inter-strand interface. Also shown is the substitution of N82 in DIX with E815 in DAX, which would eliminate the hydrogen bond with D63 of DIX and introduce electrostatic repulsion. Axin residue numbers are from the human Axin1 sequence. (**D**) Cryo-electron micrograph of DAX NQ/NG filaments.

The online version of this article includes the following figure supplement(s) for figure 4:

**Figure supplement 1.** Sequence alignment of selected Dvl and Axin DIX domains.

We sought to understand sequence differences between DIX and DAX that would contribute to their very different oligomerization behaviors. The position equivalent to N82 of DIX is DAX E815 (*Figure 4—figure supplement 1*). Given that the DIX N82D mutant destabilized the inter-strand interface, presumably due to charge-charge repulsion with D63 on the opposed strand (Fig. 2a,b), it is likely that DAX E815 would repel E794, the residue equivalent to DIX D63 (*Figure 4c*, *Figure 4—figure supplement 1*). Sequence and structure alignments revealed that relative to DIX, DAX contains a two-residue insertion, D796-C797, in the β3-β4 loop (*Schwarz-Romond et al., 2007a*; *Figure 4c*, *Figure 4—figure supplement 1*). In Dvl2, this loop (residues 60–66) packs against the opposing strand (*Figures 3b* and *4c*). The insertion, which occurs between the equivalent DIX residues 64 and 65, would therefore be expected to sterically disable formation of the inter-strand contacts needed for double helix formation. We produced a chimeric protein, designated

**Table 2.** Summary of SEC-MALS runs on Axin constructs.

| Construct | Concentration injected (μM) | Concentration at detector* (μM) | Absolute mass (kDa) | Protomers/oligomer[†] |
|---|---|---|---|---|
| DAX | 911 | 24. | 90 | 8–10 |
| DAX | 66.3 | 5.0 | 41 | 4 |
| DAX | 22.1 | 1.7 | 26 | 2–3 |
| DAX | 6.6 | 0.71 | 15 | 1–2 |
| DIDAX Y760D | 57.1 | 20. | 22 | 1 |
| DIDAX | 68.7 | 0.54 | 90–100 | 4 |

*Directly measured from maximal dRI of peak.

[†]DAX and DAX Y760D are 9.9 kDa; DIDAX and DIDAX Y760D are 26 kDa.

$DIX_{64*DC*65}$, with this insertion. Dvl2 DIX residues 60 and 61 also form part of the inter-strand interface, so we prepared $DIX_{60*DE*61}$ that contains a two-residue insertion between residues 60 and 61 that would expand this loop into and disrupt the inter-strand interface. Neither of these mutant Dvl2 DIX domains formed filaments, and each sized at roughly 2–4 monomers (**Figure 3—figure supplement 1**). While we cannot rule out the possibility that the insertion of DC after F64 disrupts the head-to-tail interactions centered on V67 and K68 (**Figure 2d**), the chromatograms show that

**Table 3.** Effect of including higher-order oligomers on estimation of DAX:DAX $K_D$ from SEC-MALS.

| Highest-order term included | $K_D$ |
|---|---|
| Dimer | 237 |
| Trimer | 672 |
| Tetramer | 844 |
| Pentamer | 907 |
| Hexamer | 932 |
| Heptamer | 941 |
| Octamer | 944 |
| Nonamer | 946 |
| Decamer | 946 |
| Undecamer | 946 |

The concentrations of each oligomeric species were determined by numerically solving the system of equations that resulted from setting the total concentration of all protomers to 710 nM, the average mass to 15 kDa, and the dissociation constants for a monomer dissociating from the end of an oligomer all equal to each other, regardless of filament length. The $K_D$ was then determined from $K_D = \frac{[\text{monomer}]^2}{[\text{dimer}]}$. An example of the system of equations for the trimer case is shown below, where m is the concentration of monomer in nM, d of dimer, and t of trimer.

The total DAX protomer concentration across all species must sum to 710 nM:

$$m + 2d + 3t = 710$$

The average mass must equal the measured 15 kD:

$$\frac{10m + 20d + 3t}{m + d + t} = 15$$

We assume the dissociation constants for a monomer dissociating from the end of an oligomer is independent of the length of the oligomer:

$$\frac{m^2}{d} = \frac{md}{t}$$

DIX$_{64*DC*65}$ runs larger than the corresponding monomer mutant, DIX$_{64*DC*65}$/Y27D, as does DIX$_{60*DE*61}$ (*Figure 3—figure supplement 1*). Combined with the presence of a distinct 'tail' (likely reflecting an equilibrium among oligomers of different length) in the DIX$_{64*DC*65}$ chromatogram that is absent in the DIX$_{64*DC*65}$/Y27D chromatogram, these data suggest that the head-to-tail interactions are at least partially intact in the DIX$_{64*DC*65}$ mutant.

Next, we used the DIX structure to predict changes that would enable human Axin1 DAX to form filaments. We changed two acidic residues to their equivalent positions in DIX (E815N/E816G), alone or in combination with changing two charged residues in the β3-β4 loop to their neutral equivalents (D793N/E794Q). These DAX mutants ('NG' or 'NQ/NG') formed double-stranded filaments as assessed by SEC and electron microscopy (*Figure 4a,d*).The E815 change appears to be critical, as the equivalent N82 in DIX forms an inter-strand hydrogen bond with D63 (*Figures 3b* and *4c*), and a negative charge introduced at N82 destabilizes double-stranded filament formation in DIX (*Figure 3a*). Overall, the data indicate that the inter-strand interactions in the double helix stabilize the DIX filament once a certain number of monomers are incorporated in a single strand, and that the DAX oligomer is small due to the inability to form inter-strand contacts.

## Binding of Axin DAX results in capped Dvl DIX oligomers

To test directly whether DAX and DIX can co-polymerize, as proposed in Axin recruitment models (*Bienz, 2014*), we mixed purified DAX with pre-formed DIX filaments in different ratios and asked whether DAX would co-sediment with DIX. Surprisingly, addition of Axin DAX reduced Dvl2 DIX filament size to the point that much of the protein no longer sedimented (*Figure 5a,b*). We confirmed these results using a total internal reflection fluorescence microscopy (TIRF) assay in which fluorescently labeled DIX filaments were mixed with unlabeled purified DAX protein. This produced a marked increase in mean fluorescence within the field of view (*Figure 5c*, *Figure 5—figure supplement 1*), presumably due to an increase in small DIX oligomers whose size is below the diffraction limit. A native gel shift assay also showed a reduction in the size of DIX oligomers upon addition of DAX (*Figure 5—figure supplement 2*). Interestingly, the DIX domain of the Wnt activator Ccd1 was also shown to disrupt Dvl DIX filaments (*Liu et al., 2011*).

To test whether the ability of DAX to disrupt DIX filaments requires insertion of DAX into one strand of a DIX filament, we repeated the sedimentation assay using DAX variants with mutations in their head (Y760D) or tail (V800A/F801A) that block the intra-strand interaction (*Fiedler et al., 2011*). Both mutants were able to solubilize the DIX filaments (*Figure 5b*). However, DAX mutated in both the head and tail interfaces, Y760D/V800A/F801A, failed to solubilize the DIX filament (*Figure 5b*). These results indicate that Axin DAX solubilizes Dvl DIX via its head-to-tail interface. With an input concentration of 10 μM DIX, the half-maximal loss of sedimentation occurred at approximately 15 μM DAX (*Figure 5b*). This concentration dependence was also observed in the native gel shift experiment (*Figure 5—figure supplement 2*). These data indicate that the DAX-DIX interaction is at best comparable to the homotypic DIX-DIX interaction, and weaker than the DAX-DAX interaction, observations consistent with previous NMR analyses (*Fiedler et al., 2011*).

The ability of the head or tail mutants to solubilize DIX filaments suggests that DAX does so by capping (*Figure 5d*). FRAP measurements of Dvl-GFP expressed in cells indicate that Dvl polymers are highly dynamic (*Schwarz-Romond et al., 2007a*; *Schwarz-Romond et al., 2005*; *Schwarz-Romond et al., 2007b*). These findings suggest that DAX can incorporate into a dynamic DIX filament by its head-to-tail interface (*Figure 5d*). Because DAX cannot form the inter-strand interaction, this locally destabilizes the DIX filament, resulting in severing of large filaments in vitro (*Figure 5d*). The fact that approximately 40% of the DIX still sediments even with a fivefold molar excess of DAX is likely due in large part to the relatively weak DIX-DAX K$_D$ of 10–20 μM (*Yamanishi et al., 2019a*). Furthermore, DAX insertions near the ends of DIX filaments would consume DAX while only shifting a small portion of the DIX contained in that filament from the pellet to the supernatant. Likewise, DAX insertion into a DIX oligomer that is already too small to sediment (the likelihood of which would increase as the number of severed DIX oligomers increased) would consume DAX without shifting any DIX from the pellet to the soluble fraction.

The role of oligomer stability in determining the DAX-DIX interaction was examined further by assessing the role of a continuous electronegative groove that runs along the DIX filament (*Figure 5—figure supplement 3a*). The acidic residues that form this groove are strongly conserved in most Dvl proteins, but not Axin (*Figure 4—figure supplement 1*). We hypothesized that this feature

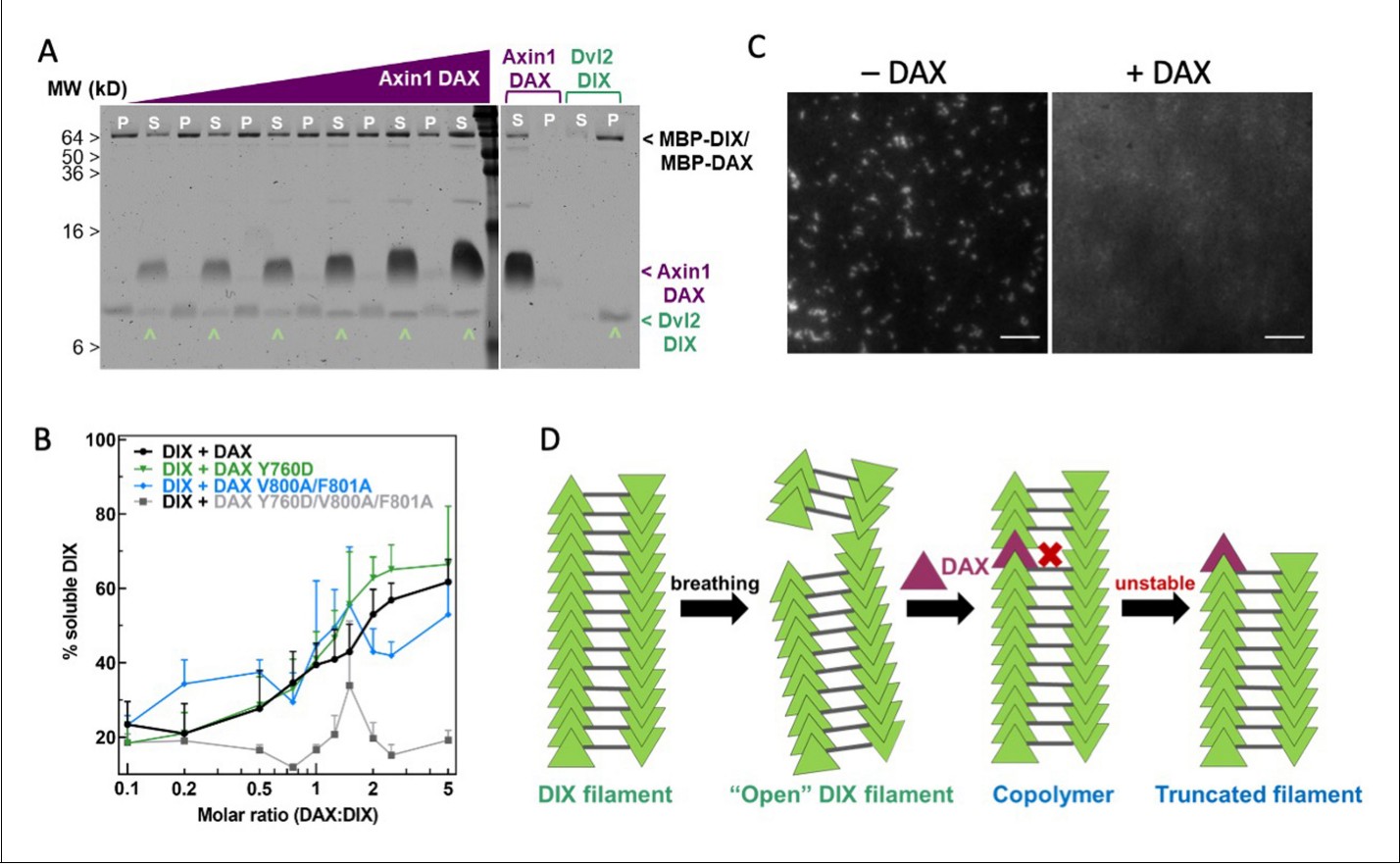

**Figure 5.** Axin1 DAX solubilizes Dvl2 DIX filaments. (**A**) Sedimentation of preformed DIX filaments (10 µM) mixed with increasing amounts of purified wild-type DAX, showing shift of DIX from the pellet (P) into the supernatant (S) (denoted byˆ). A representative SDS-PAGE gel (with a portion excised for clarity) is shown here. (**B**) Quantification of solubilization of DIX filaments by wild-type DAX (black curve) and DAX with mutations in the head interface (Y760D; green curve), tail interface (V800A/F801A; blue curve), or both (Y760D/V800A/F801A; gray curve) carried out as in (**A**). (**C**) Representative images of 10 µM fluorescently labeled DIX filaments in the absence and presence of 50 µM purified DAX, showing increase in background fluorescence associated with filament disruption. Scale bar = 5 µm. (**D**) Model for DAX incorporation into a filamentous DIX oligomer in vitro.

The online version of this article includes the following figure supplement(s) for figure 5:

**Figure supplement 1.** Change in mean fluorescence in TIRF DIX filament assay in the absence and presence of DAX.

**Figure supplement 2.** Native gel analysis of DIX filament mobility in the presence of DAX.

**Figure supplement 3.** A conserved electronegative groove in the Dvl2 DIX double helix enables association with Axin1 DAX.

is important to Dvl2 function and mutated two of the acidic residues, E22 and E24, to either glutamines ('QQ') or lysines ('KK'), to reduce the negatively charged character of the groove. These residues do not contribute to the head-to-tail interaction in the DIX filament structure, and both the QQ (*Figure 5—figure supplement 3b*) and KK mutants (data not shown) formed double helical filaments. Remarkably, neither mutant could be solubilized by DAX in the sedimentation assay (*Figure 5—figure supplement 3c*). Also, the mutants showed nearly 100% sedimentation, vs. 80% in the WT case (*Figure 5—figure supplement 3c*), suggesting that they are more stable. Both pairs of mutations significantly impaired Dvl2 signaling (*Figure 5—figure supplement 3d*). We hypothesize that the acidic residues tune DIX oligomer stability such that it allows incorporation of Axin DAX; the QQ or KK mutations hyperstabilize the double stranded oligomer and thereby prevent binding of Axin indirectly by making the energetic penalty for incorporation much higher than in wild-type DIX. We cannot rule out that the mutations directly affect DAX affinity for DIX, as a recent structure of a non-filament forming DIX-DAX heterodimer revealed that E24 interacts with K789 on DAX (*Yamanishi et al., 2019a*). In this scenario, replacing E24 with lysine would be expected to disrupt this interaction, but the glutamine substitution would be tolerated. However, as the QQ and KK

mutants display significant and identical TOPFLASH signals (*Figure 5—figure supplement 3d*), it seems more likely that the E24-K789 interaction does not contribute significantly to the DIX-DAX interaction.

## Avidity enables Axin recruitment by Dvl2 to drive signaling

We hypothesized that the DIX oligomer produced by double strand formation provides avidity (*i.e.*, multiple binding sites) that compensates for the intrinsically weak DAX-DIX interaction. To test this hypothesis, we assayed the ability of Dvl2 constructs bearing varying degrees of affinity and avidity for Axin binding to restore TOPFLASH signaling in Dvl TKO cells. The DIX insertion mutants $DIX_{60*DE*61}$ and $DIX_{64*DC*65}$ that disrupted the inter-strand interface (*Figure 3—figure supplement 1*) also ablated signaling (*Figure 6*). Based on this observation, we expected that Dvl2 with its DIX domain replaced by Axin1 DAX, which likewise cannot form double-stranded oligomers, would behave similarly. Remarkably, however, the DAX swap more than quadrupled the signal relative to wild-type DIX (*Figure 6*). We suggest that this result reflects the higher affinity of the DAX-DAX

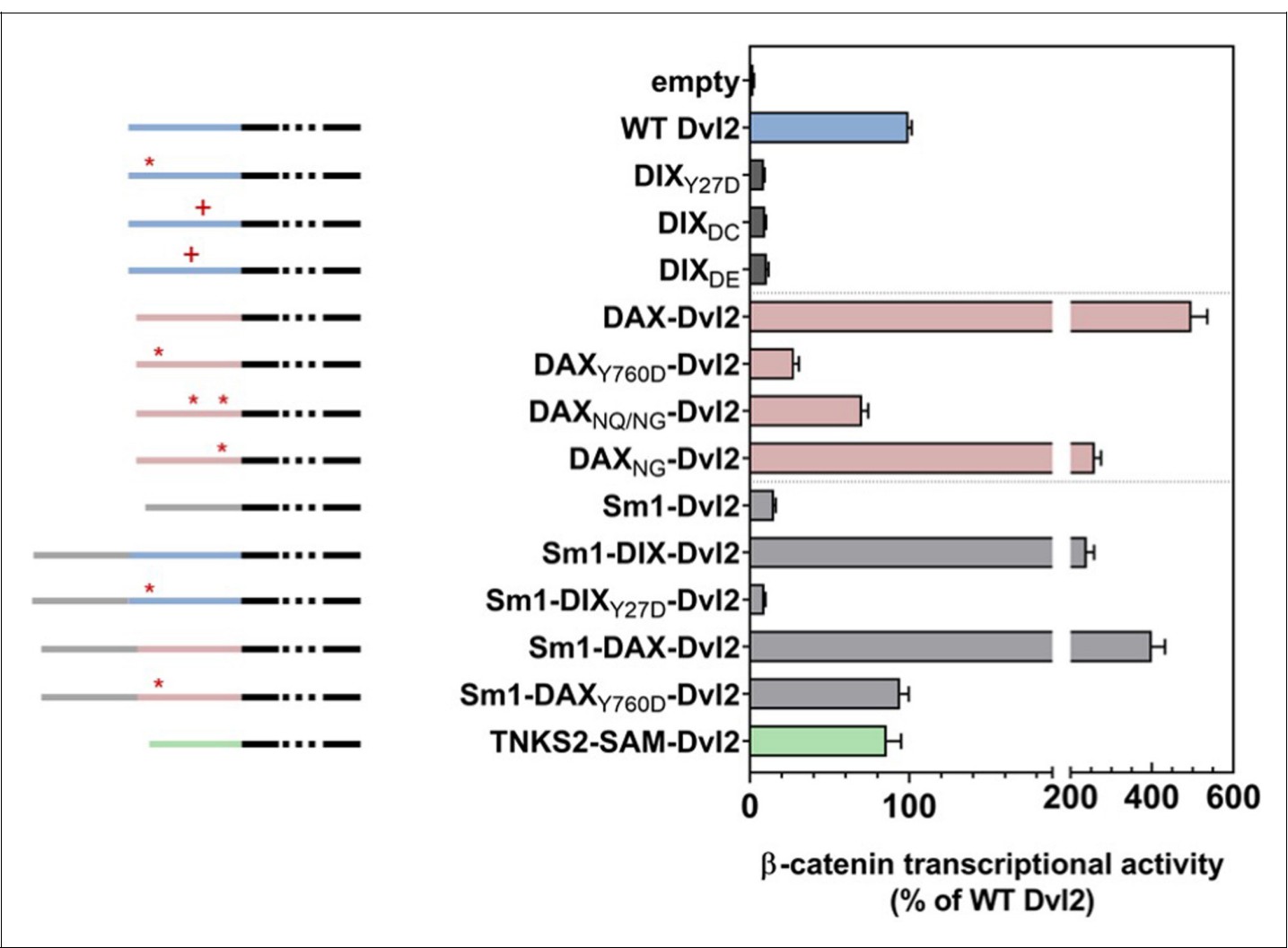

**Figure 6.** Wnt-dependent β-catenin signaling by Dvl2 constructs containing. DIX mutants affecting either the intra- (Y27D) or inter- (DC, DE) strand interactions (dark gray); replacement of DIX with wild-type or mutant DAX domains (pink); or addition of heterologous oligomerization domains (Sm1, light gray; Tankrase2 SAM, green) expressed at near-endogenous levels in Dvl triple knockout (TKO) HEK293 cells, as measured by TOPFLASH. Schematic diagrams of the constructs are shown to the left: the portion of Dvl C-terminal to the DIX domain is shown in black, the Dvl DIX domain in blue, the Axin1 DAX domain in pink, the Sm1 heptamerization domain in light gray, and the Tankyrase2 SAM domain in green. The red asterisks indicate 1) the DIX Y27D or the equivalent DAX Y760D head mutant that blocks oligomerization, or 2) mutated sites within DAX that increase filament-forming propensity ($DAX_{NQ/NG}$, $DAX_{NG}$). Red pluses indicate residue insertions (DC, DE) that interfere with the DIX inter-strand interface.

interaction, which can compensate for the loss of avidity resulting from the formation of single-stranded oligomers. This idea is consistent with several observations. First, replacing the Dvl2 DIX domain with the DAX NG or NQ/NG variants, which can form double-stranded oligomers, reduces signaling relative to the WT DAX swap (*Figure 6*). Also, the hyperstabilized Dvl2 DIX QQ and KK mutants reduce signaling relative to WT DIX (*Figure 5—figure supplement 3*). These data indicate that there is a balance between the avidity provided by the double stranded oligomer and the energetic penalty that it produces for binding WT DAX.

We further probed the role of avidity for Axin by testing the ability of Dvl2 constructs containing a ring-shaped heptameric protein, Sm1 from *Archaeoglobus fulgidus* (*Törö et al., 2002*), to restore TOPFLASH signaling (*Figure 6*). Dvl2 with Sm1 but lacking a DIX domain did not restore signaling, nor did a construct with Sm1 and the non-polymerizing Y27D DIX domain, which can bind DAX (*Yamanishi et al., 2019a*; *Figure 6*). In contrast, Dvl2 containing both Sm1 and a wild-type DIX domain signaled at ~3 fold higher levels than wild-type Dvl2. Thus, the presence of the Sm1 heptamer combined with native DIX enhances signaling, which we interpret as an enhancement of avidity for Axin.

We next tested the combination of Sm1 with the DAX domain in Dvl2 but found no significant enhancement of signaling (*Figure 6*), suggesting that the native DAX-DAX interaction provides sufficiently high affinity to recruit Axin to the receptor complex. However, while replacing the Dvl2 DIX domain with the non-polymerizing Axin1 DAX Y760D significantly reduced signaling, combining Sm1 with DAX Y760D restored signaling to the same level as that of wild-type Dvl2 (*Figure 6*). As Y760D can only form a single DAX-DAX interaction with endogenous Axin, this result indicates that Sm1 oligomerization provides a means to recruit a sufficient number of Axin molecules for wild-type level signaling.

The correlation of signaling with improved Axin recruitment is also supported by the replacement of Dvl2 DIX with the Tankyrase2 SAM domain (*Figure 6*). Like DIX domains, SAM domains form head-to-tail assemblies (*Bienz, 2014*). Although the SAM domain does not bind to Dvl or Axin, it does oligomerize Tankrase2, which in turn binds Axin (*Mariotti et al., 2016*). That the Tankyrase 2 SAM replacement restores signaling likely indicates that the SAM domain present in the Dvl2 construct indirectly recruit Axin via endogenous Tankyrases.

## The size of Dvl oligomers is limited in cells

When Dvl or Axin fused to a fluorescent protein is overexpressed, activation of Wnt signaling produces membrane-proximal puncta containing Dvl and Axin (e.g., *Cliffe et al., 2003*; *Pronobis et al., 2017*; *Schwarz-Romond et al., 2007b*). The ability of Dvl or Axin to form puncta correlates with the ability of purified DIX domains to oligomerize, and in the case of Dvl DIX, form filaments. Moreover, Dvl DIX polymerization mutants prevent signaling (*Fiedler et al., 2011*; *Schwarz-Romond et al., 2007a*; *Figures 3c* and *5*), indicating that oligomerization of Dvl is necessary for signal transduction. However, the extent to which Dvl oligomerizes at endogenous levels has not been established (*Gammons et al., 2016b*; *Ma et al., 2020*; *Schaefer et al., 2018*; *Smalley et al., 2005*). Using HEK293T cells in which a C-terminal GFP fusion was knocked into the endogenous Dvl2 locus, we performed live epifluorescent imaging with and without Wnt3A stimulation. Although we confirmed that the cell line expressed Dvl2 at the same level as wild-type HEK293T cells and responded to Wnt3a stimulation (*Figure 7—figure supplement 1*), we failed to observe the multiple large Dvl clusters seen in overexpression studies. Irrespective of Wnt 3A stimulation, most cells had diffuse signal; ~20% had 1–2 observable and stable puncta, and ~2% had more than two such puncta (see Materials and methods-Live cell epifluorescene imaging; *Video 1*). These observations confirm recent findings that Dvl at endogenous expression levels showed few large clusters other than the

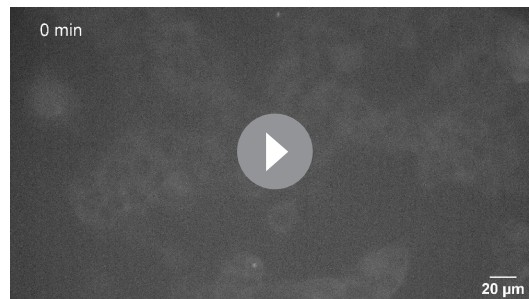

**Video 1.** Dvl2-GFP long-term epifluorescent live cell imaging. HEK293T Dvl2-GFP/Axin1-RFP cells were imaged in HEPES buffered media for ~15 hr.
https://elifesciences.org/articles/55015#video1

likely MTOC/centrosome associated puncta (*Gammons et al., 2016b*; *Ma et al., 2020*).

Wnt-activated Fzd has been thought to recruit Dvl to the membrane and activate Dvl oligomerization as the first step in signalosome formation (*Gammons et al., 2016a*; *Gammons et al., 2016b*; *Nusse and Clevers, 2017*). We therefore utilized TIRF to image Dvl2-GFP near the membrane to achieve high spatiotemporal resolution with high signal-to-noise ratio. To avoid issues of ligand accessibility, we developed an apical TIRF imaging setup (*Jaykumar et al., 2016*) in which the cells grown on PET filters are inverted with their apical sides down, with or without Wnt3A stimulation (*Figure 7a,b*). In both cases we observed many diffraction-limited low intensity spots per cell, indicative of complexes containing small numbers of GFPs. Approximately 10% of cells displayed 1–2 large puncta, some of which may be MTOC-associated Dvl, and rarely displayed more than two large puncta. Some of the diffraction-limited spots exhibited step photobleaching corresponding to single GFP photobleaching events. We performed single-particle tracking (*Jaqaman et al., 2008*), inferred the approximate single GFP intensity using Gaussian mixture models fitted to intensity distributions, and found that 90% of the spot intensities were within 10x of a single GFP intensity with or without Wnt3A stimulation (*Figure 7c*). Moreover, we could not detect a significant difference in the number of Dvl2-GFP molecules in these spots upon Wnt3A stimulation. Our results strongly suggest that at endogenous levels, membrane associated Dvl2-containing complexes rarely contain more than about 10 Dvl2 molecules, with or without Wnt3a stimulation.

## Discussion

Our data indicate that the Dvl DIX oligomer must have a sufficient number of monomers to form the double stranded structure found in the filaments: disrupting the inter-strand interface reduces signaling, implying that the double stranded DIX oligomer structure is functionally relevant. Although the formation of DIX filaments in vitro reflects fundamental properties of DIX-DIX interactions, the TIRF imaging data (*Figure 7*) indicate that a typical Dvl oligomer contains fewer than 10 molecules. We do not know the smallest stable double stranded oligomeric unit of DIX. The more severe effects of the head-to-tail interface mutations on size (*Figure 3a*, *Figure 3—figure supplement 1*) and signaling (*Figure 3c*) suggest that there must be at least 4 DIX protomers (i.e., a pair of head-tail dimers) to form a stable double-stranded structure.

The in vitro DIX filament severing activity of Axin DAX, combined with the inability of DAX to form the antiparallel inter-strand interface found in DIX filaments, implies that DAX binds to the ends or 'caps' the end of a double-stranded DIX oligomer (*Figures 5d* and *8*). Binding to each end of each strand would result in at most 4 Axin protomers associated with a given Dvl multimer. The TIRF data (*Figure 7c*), however, may not distinguish binding of Axin to the ends of a short, pre-existing DIX oligomer versus severing of small oligomers (*Figure 5d*) in a subpopulation of molecules. These scenarios are not mutually exclusive. Severing would be consistent with the weakened signaling of the E22Q/E24Q and E22K/E24K mutants, which appear to form hyperstabilized filaments (*Figure 5—figure supplement 3*). Severing might also explain the observation that mild overexpression of Dvl did not significantly affect wingless signaling (*Schaefer et al., 2018*). Crucially, capping and/or severing are distinct from models in which the Axin interaction with Dvl occurs by co-polymerization of DAX and DIX (*Bienz, 2014*), which would not limit the number of Axin molecules associated with Dvl.

Wnt binding results in recruitment of Axin to the receptor complex in a Dvl-dependent manner (*Cliffe et al., 2003*), and prior models have postulated that Wnt binding to Fzd and LRP5/6 promotes Dvl oligomerization needed for Axin binding (*Gammons et al., 2016a*; *Gammons et al., 2016b*; *Nusse and Clevers, 2017*). However, our TIRF data (*Figure 7c*), as well as those of *Ma et al., 2020*, indicate that there is not a major change in the oligomeric state of Dvl at the membrane upon Wnt activation. Thus, how Wnt binding to its receptors 'activates' Dvl for Axin recruitment remains unclear. Although we did not detect a statistically significant effect of Wnt stimulation on Dvl2 oligomer size, we do not rule out a change in oligomeric state that retains the same total number of Dvl molecules in a diffraction-limited spot. We also cannot formally rule out that a much larger Dvl oligomer forms transiently upon Wnt signaling and is subsequently severed by Axin. Improved imaging with high time resolution may allow us to address these possibilities.

The DIX-DIX interaction has a $K_D$ in the 5–20 µM range (*Schwarz-Romond et al., 2007a*; *Yamanishi et al., 2019a*), and our data are consistent with this estimate (*Figure 1c,d*). Attempts to

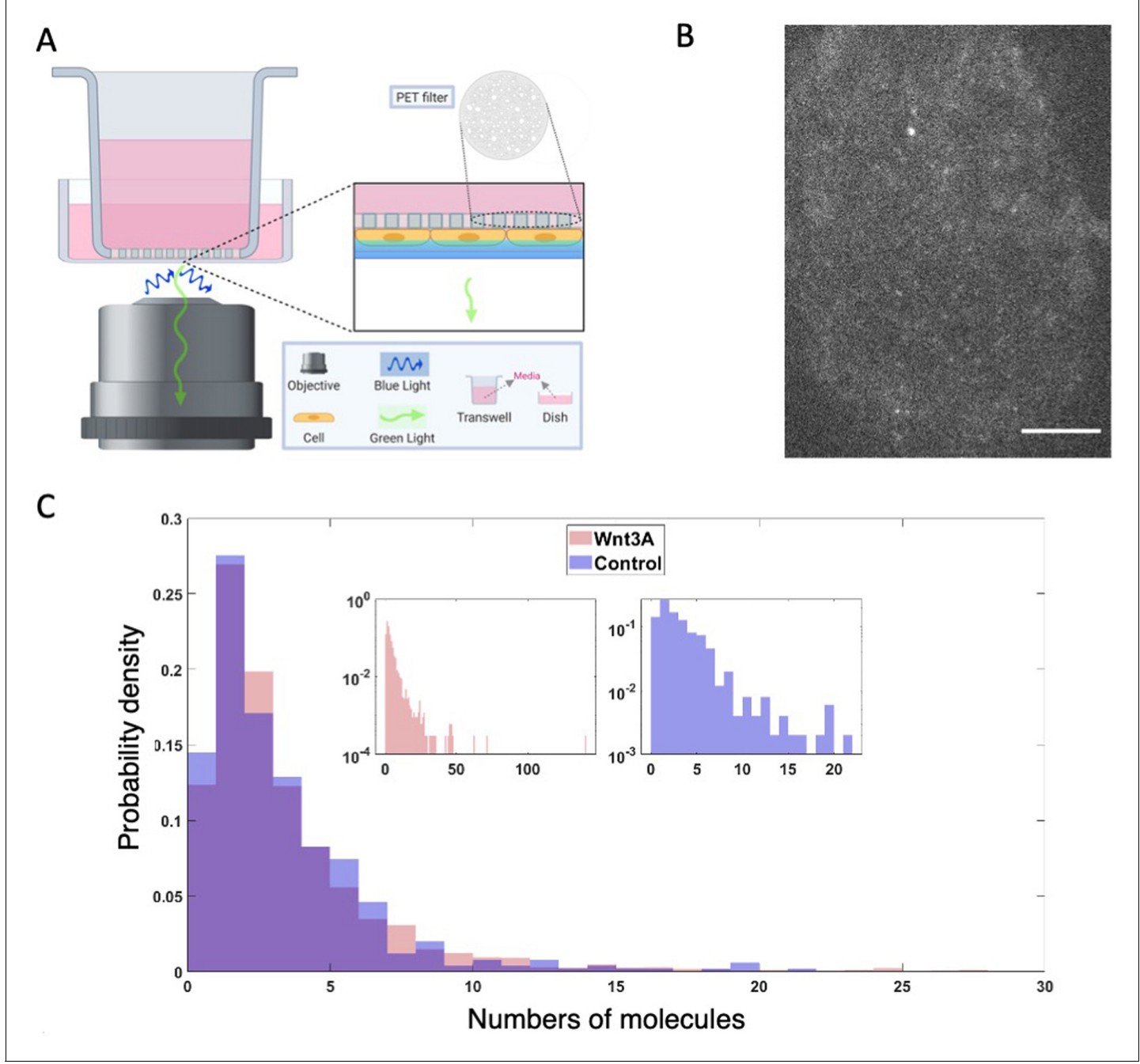

**Figure 7.** Dvl2 forms small oligomers in cells. (**A**) Setup for live cell TIRF imaging of the apical membrane of HEK293T Dvl2-GFP/Axin1-dsRed knockin cells. (**B**) Representative TIRF image. Scale bar = 5 μm. (**C**) Distribution of measured spot intensities shown as the average number of GFPs equivalent to measured value.

The online version of this article includes the following figure supplement(s) for figure 7:

**Figure supplement 1.** Characterization of Dvl2-GFP/Axin1-dsRed knockin HEK293T cells.

measure DAX-DAX or DAX-DIX affinities have employed head-to-tail mutants with the goal of isolating the affinity of a single interface. These experiments have suggested $K_D$ values much weaker than that of DIX-DIX (*Fiedler et al., 2011*; *Schwarz-Romond et al., 2007a*; *Yamanishi et al., 2019a*). The filament disruption experiments (*Figure 5*, *Figure 5—figure supplement 1*, *Figure 5—figure supplement 2*) indicate that the affinity of the heterotypic DIX-DAX interaction is on the same order as that of DIX-DIX. However, our SEC-MALS data indicate that the DAX-DAX interaction is roughly an

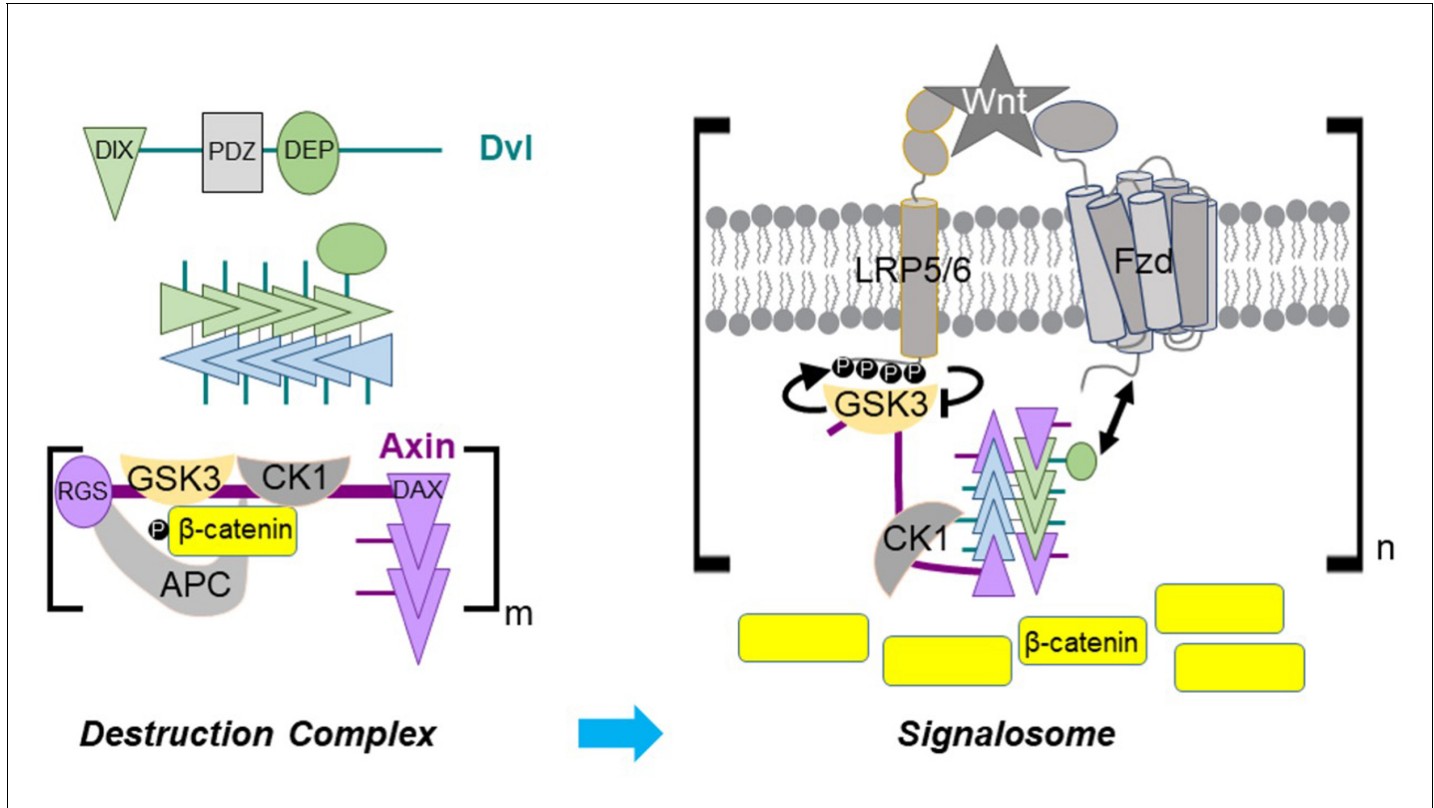

**Figure 8.** Dvl DIX oligomerization in Axin recruitment upon Wnt signaling. (Upper left) Schematic diagram of Dvl primary structure and association of Dvl DIX into an oligomer. A Dvl DIX oligomer provides up to four binding sites for Axin, one at each end of each paired strand. (Lower left) In the absence of Wnt signaling, self-association of the Axin DAX domain and multivalent interactions of Axin with APC create a crosslinked, possibly phase separated, destruction complex with multiple copies of the constituent proteins. (Right) The Dvl DEP domain (green oval) binds to Fzd and the Dvl oligomer associates with up to 4 Axin DAX domains. GSK-3 bound to Axin generates its own inhibitor by phosphorylating the LRP5/6 cytoplasmic tail. Axin DAX binding to the ends of the DIX oligomer provides an optimal relative stoichiometry of GSK-3–bound Axin and LRP5/6 to enable efficient inhibition of GSK-3. Some bound Axin molecules as well as DEP domains from the Dvl oligomer may interact with other receptor complexes, thereby producing a crosslinked signalosome.

order of magnitude stronger than that of DIX-DIX or DIX-DAX (*Figure 4b*, *Table 2*, *Table 3*). The mutants used in the other studies of DAX-DAX interactions may have effects beyond the presumed simple disruption of the head-to-tail interface; for example, interfaces may be energetically coupled, rather than independent of one another as is assumed in the mutational experiments.

The β-catenin destruction complex is thought to be assembled by oligomerization of Axin through its DAX domain and binding of Axin to APC. The complex may be a phase-separated structure that enables efficient β-catenin destruction due to the high local concentration of its enzymes and substrates (*Schaefer and Peifer, 2019*). Scaffold proteins with multiple binding sites for each other can crosslink to form phase separated structures; if one component is in excess, however, the crosslinking probability is diminished and phase separation does not occur (*Banani et al., 2016*). Imaging of Axin and APC overexpressed in mammalian cells has revealed roughly equal amounts of the two proteins in destruction complex puncta (*Schaefer et al., 2018*). Given that APCs contain multiple Axin-binding SAMP repeats and a conserved self-association region (*Kunttas-Tatli et al., 2014*), the limited DAX oligomerization observed in vitro seems well-matched to the number of Axin binding sites in APC, consistent with the proposed phase separation. The ability of β-catenin to bind to both APC and Axin simultaneously (*Ha et al., 2004*) also introduces a third component potentially capable of crosslinking these structures. Axin is present at about 150 nM in HEK293T cells (*Tan et al., 2012*), which would predict no DAX oligomerization based on our estimated $K_D$ for DAX-DAX association. However, the ability of other regions of Axin and APC to interact, as well as the effects of the crowded cellular environment, may enhance the effective DAX-DAX affinity. The

amount of Axin in endogenous destruction complex puncta in *Drosophila* embryos is estimated to be in the 10's-100's of molecules (*Schaefer et al., 2018*). An in vivo analysis in which Axin was over-expressed approximately 4x indicated that DAX is not absolutely essential for destruction complex function (*Peterson-Nedry et al., 2008*); perhaps the multiple Axin-binding sites in APC, combined with the elevated concentration of Axin and the ability of β-catenin to bind both Axin and APC, provided sufficient crosslinking to enable destruction complex function in that case. Also, in the absence of Wnt signal, overexpression of Axin or APC has little effect on β-catenin destruction (*Schaefer et al., 2018*), which may indicate that the system is sufficiently efficient that increasing either component beyond normal level has no significant effect on destruction.

Although the DAX domain may not be absolutely necessary for destruction complex function in vivo, it is essential for turning off destruction (*Peterson-Nedry et al., 2008*), highlighting the importance of the Axin-Dvl interaction. The stronger DAX-DAX homotypic interaction compared to DIX-DAX may prevent disruption of the destruction complex by cytoplasmic Dvl in the absence of a Wnt signal. Dvl DIX oligomerization likely confers increased avidity on Dvl binding to Axin, which would be needed to overcome the stronger homotypic DAX-DAX interaction and thereby recruit the destruction complex to the activated receptors. This is consistent with the observation that modest (~3x) overexpression of Axin can be tolerated during fly development (*Peterson-Nedry et al., 2008*; *Schaefer et al., 2018*; *Wang et al., 2016*), whereas higher (8-9x) levels will turn off signaling (*Cliffe et al., 2003*; *Schaefer et al., 2018*; *Willert et al., 1999*). Upon signaling, activation of Dvl results in the high-avidity binding of DAX to DIX needed to effectively recruit the destruction complex to the receptor complex and disrupt the function of the destruction complex (*Figure 8*). Such a model explains the observation that substituting DAX for DIX in Dvl greatly enhances signaling (*Figure 6*), as the higher self-affinity of Axin DAX would more readily recruit Axin to the receptor complex. Conversely, the loop insertion mutants that disrupt DIX oligomerization, and the inter-strand Dvl DIX mutants, which do not oligomerize to the same extent as wild-type DIX, may reduce signaling by lowering the effective avidity for Axin, although we cannot rule out that they directly impair the Axin-binding interface, particularly the loop insertions (*Figure 6*). The observation that the Sm1 heptamer can rescue signaling in a non-oligomerizing Dvl2 mutant (Sm1-DAX$_{Y760D}$-Dvl2), which can still form a single head-to-tail heterodimer with Axin, also supports the notion that avidity has an important role in recruiting Axin to the receptor complex. The equivalent fusion with DIX Y27D does not rescue, which can be attributed to the much weaker DIX-DAX interaction.

The limit of four Axin-binding sites per activated Dvl oligomer in a single receptor complex may provide an optimal stoichiometry for inhibition of GSK-3 (*Figure 8*). The cellular concentrations of Axin and GSK-3 have been reported to lie in the tens–hundreds of nanomolar range (*Tan et al., 2012*). We have measured the affinity of GSK-3 and Axin to be $K_D$ = 8 nM (MDE and WIW, manuscript in preparation), implying that essentially all Axin brought to the activated receptor complex will be bound to GSK-3. A single phosphorylated LRP6 motif can inhibit GSK-3 with a $K_i$ of 2 µM (*Stamos et al., 2014*). Five inhibitory motifs in the LRP5/6 tail cooperate in Wnt/β-catenin signal transduction (*MacDonald et al., 2008*; *Wolf et al., 2008*), and it is not known how many GSK-3 molecules can be stably inhibited by one phosphorylated tail. Multiple motifs could be needed to ensure high rebinding probability of GSK-3 to achieve highly effective inhibition; alternatively, each motif could inhibit a GSK-3 molecule. In any case, there cannot be a great excess of GSK-3 relative to LRP5/6 for efficient inhibition of GSK-3. Therefore, a limited number of Axin/GSK-3 complexes associated with a single Fzd-LRP5/6 complex could maximize the GSK-3 inhibitory activity of an activated receptor complex.

The preceding analysis considered the relative stoichiometry of a single LRP5/6–Fzd/Dvl complex with Axin-GSK-3. However, multiple phosphorylated LRP5/6 molecules may be needed to stabilize enough β-catenin molecules to trigger target gene activation. It has been suggested that the activated 'signalosome' is a crosslinked, perhaps phase separated, structure that would include multiple copies of the receptors (*Gammons and Bienz, 2018*; *Schaefer and Peifer, 2019*). The presence of <10 Dvl molecules in the puncta observed in our and other (*Ma et al., 2020*) TIRF experiments, and the binding of Axin DAX to the ends of Dvl DIX oligomers, indicates that there are likely to be roughly matched numbers of Axin and Dvl associated with the activated receptors. This would provide the appropriate relative stoichiometry to optimize crosslinking needed for a functional signalosome. It is important to note, however, that understanding the precise connection of the properties of the isolated DIX and DAX domains to the behavior of Dvl and Axin will require experiments using

purified, full-length proteins. For example, both proteins have large unstructured regions and are also regulated by post-translational modifications, features that could contribute to phase separation behavior (*Alberti et al., 2019*; *Feng et al., 2019*; *Owen and Shewmaker, 2019*; *Snead and Gladfelter, 2019*). Moreover, it is possible that other regions of the Dvl protein affect the extent of DIX domain oligomerization, and likewise there may be intramolecular regulation of the Axin DAX domain (*Kim et al., 2013*). Finally, 10's-100's of Axin molecules were found in puncta present in Wnt stimulated *Drosophila* embryo cells when Axin was expressed at about 4x endogenous levels (*Schaefer et al., 2018*). The reason for the difference with our measurements is not clear, but it may indicate a significant concentration dependence of oligomerization, or perhaps that different regulatory mechanisms operate in the fly and mammalian systems. Moving forward, it will be essential to determine experimentally the relative numbers and regulatory states of receptors, Dvl and Axin in a signalosome.

# Materials and methods

## Key resources table

| Reagent type (species) or resource | Designation | Source or reference | Identifiers | Additional information |
|---|---|---|---|---|
| Strain, strain background (*Escherichia coli*) | BL21 (DE3) Codon-Plus RIL | Agilent | 230245 | Strain for expressing recombinant proteins |
| Strain, strain background (*Escherichia coli*) | XL10-Gold ultracompetent cells | Agilent | 200314 | Strain used for molecular biology and creating recombinant DNA |
| Cell line (*Homo-sapiens*) | Dvl TKO HEK293T | Stephane Angers, University of Toronto See *Gammons et al., 2016b* https://doi.org/10.1242/jcs.195685 | | CRISPR/Cas9 deletion of hDvl1, hDvl2, and hDvl3 *Authentication methods:* deletions were confirmed by genomic DNA sequencing; immunoblotting with anti-Dvl2 (*Figure 7—figure supplement 1*) *Mycoplasma contamination testing status:* Tested |
| Cell line (*Homo-sapiens*) | Dvl2-GFP/dsRed-Axin1 HEK293T | This paper; cells were derived from cells purchased from the European Collection of Cell Cultures | | C-terminal tagging by CRISPR/Cas9 gene editing *Authentication methods:* STR DNA profiling; flow cytometry and fluorescence microscopy using GFP/RFP; local genomic DNA sequencing; and anti-Dvl2 (*Figure 7—figure supplement 1*) and anti-Axin1 immunoblots *Mycoplasma contamination testing status:* Tested Contact Bienz laboratory for distribution |
| Cell line (mouse) | L and L3A | ATCC | L CRL-2648 L3A CRL-2647 | Used for generating control and Wnt-conditioned media; See https://web.stanford.edu/group/nusselab/cgi-bin/wnt/ |
| Recombinant DNA reagent | pCS2+ | This paper | | Ampicillin resistance; expression in mammalian cell culture; includes a N-terminal M2 Flag tag Contact Weis lab for distribution |

*Continued on next page*

*Continued*

| Reagent type (species) or resource | Designation | Source or reference | Identifiers | Additional information |
|---|---|---|---|---|
| Recombinant DNA reagent | pCDF-Duet-His6-MBP-TEV | Novagen; modified in Weis lab to include MBP-TEV | 71340 | Streptomycin resistance; expression in bacterial cultures Contact Weis lab for distribution |
| Recombinant DNA reagent | pGEX-TEV | *Choi et al., 2006* https://doi.org/10.1074/jbc.M511338200 | | Ampiicillin resistance; expression in bacterial cultures; pGEX-KG plasmid (ATCC) with a new TEV protease site Contact Weis lab for distribution |
| Chemical compound, drug | Fetal Bovine Serum | Gemini GemCell, U.S. Origin | | Used to generating control and Wnt-conditioned media; provides low basal activity |
| Chemical compound, drug | LB Broth, Miller, granules | Fisher BioReagents | | |
| Chemical compound, drug | Amylose agarose | New England BioLabs | | For purification by MBP affinity |
| Chemical compound, drug | Glutathione agarose | Pierce | | For purification by GST affinity |
| Chemical compound, drug | Negative stain grids (carbon-coated copper) | EMS | CF200-Cu | |
| Chemical compound, drug | Cryo-EM lacey grids | EMS | LC200-Cu | Freezing done with Leica EM GP |
| Antibody | Dvl2 polyclonal antibody | Cell Signaling Technology | 3216 | IB (1:1000) RRID:AB_2093338 |
| Antibody | Axin1 antibody | Cell Signaling Technology | C76H11 | IB (1:1000) RRID:AB_2054638 |
| Antibody | Beta-catenin monoclonal antibody L54E2, AlexaFluor647-conjugated | Cell Signaling Technology | 4627 | IF (1:300) RRID:AB_10691326 |
| Recombinant DNA reagent | Dvl2 (mouse) | This paper | GenBank U24160.2 | residues 2–736; DIX = 12–92; Plasmid in pCS2+-M2-Flag; contact Weis lab for distribution |
| Recombinant DNA reagent | Axin1 (human) | This paper | NCBI NM_181050.3 | corresponds to residues 1–826 of NP_851393.1 DAX = 743–826 DI-DAX = 599–826 Plasmid in pCAN-M2S-myc; contact Weis lab for distribution |
| Recombinant DNA reagent | Tankyrase2 SAM domain (human) | Nai-Wen Chi (Addgene plasmid # 34691) | NCBI NP_079511.1 | residues 867–940 |
| Recombinant DNA reagent | Sm1 residues 1–77 heptamerization domain (archaea) | Integrated DNA Technologies/This paper | NCBI WP_010878376.1 | Contact Weis lab for distribution PDB: 1LJO; *Törö et al., 2002* |
| Commercial assay or kit | MALS | Wyatt | | See main text for more details |
| Commercial assay or kit | S75, S200, Superose 6 10/300 | Pharmacia/GE | | 24 mL 'increase' columns are tolerant of high flow rates and have slight differences in elution profile |

*Continued on next page*

*Continued*

| Reagent type (species) or resource | Designation | Source or reference | Identifiers | Additional information |
|---|---|---|---|---|
| Commercial assay or kit | TopFlash Dual-Light Reporter Gene Assay System | ThermoFisher/Applied Biosystems | T1005 | |
| Commercial assay or kit | AlexaFluor-488/ 647 C2 maleimide | ThermoFisher/Molecular Probes | | See Materials and methods for more details |
| Commercial assay or kit | Phusion HiFi DNA polymerase | Fermentas/ Thermo Fisher | | |
| Commercial assay or kit | FastDigest Restriction endonucleases | Fermentas/ Thermo Fisher | | |
| Commercial assay or kit | Gibson Assembly HiFi 1-Step Kit | SGI DNA | GA1100-10 | |
| Commercial assay or kit | Stain-free TGX | Biorad | | Specifically visualizes Trp-containing proteins, using Gel Doc EZ |
| Commercial assay or kit | Any Kd TGX | Biorad | 4569036 | Used for native gel runs |
| Commercial assay or kit | LiCOR IR-dye secondary antibody and Odyssey 3.0 imaging system | LiCOR, Inc | | Used for visualizing immunoblots and quantifying sedimentation assays |
| Software, algorithm | RELION 3.0.8 | *He and Scheres, 2017* https://doi.org/ 10.1016/j.jsb.2017.02.003 *Scheres, 2012* https://doi.org/10.1016/ j.jmb.2011.11.010 *Zivanov et al., 2018* https://doi.org/10.7554/ eLife.42166 | RRID:SCR_016274 | |
| Software, algorithm | CTFFIND-4.1 | *Rohou and Grigorieff, 2015* https://doi.org/10.1016/ j.jsb.2015.08.008 | RRID:SCR_016732 | |
| Software, algorithm | Phenix | *Afonine et al., 2018* https://doi.org/10.1107/ S2059798318006551 | RRID:SCR_014224 | |
| Software, algorithm | Coot | *Emsley et al., 2010* https://doi.org/10.1107/ S0907444910007493 | RRID:SCR_014222 | |
| Software, algorithm | FIJI | *Schindelin et al., 2012* https://doi.org/ 10.1038/nmeth.2019 | RRID:SCR_002285 | |
| Software, algorithm | Astra 6 | Wyatt Technologies | RRID:SCR_016255 | |
| Software, algorithm | SBGrid | *Morin et al., 2013* https://doi.org/ 10.7554/eLife.01456 | RRID:SCR_003511 | |
| Software, algorithm | UCSF Chimera | *Pettersen et al., 2004* https://doi.org/ 10.1002/jcc.20084 | RRID:SCR_004097 | v1.14 |
| Software, algorithm | PyMOL | Schrödinger, LLC | RRID:SCR_000305 | v2.3.3 |
| Software, algorithm | GraphPad Prism 8.0.2 | GraphPad Software, Inc | Version 263 RRID:SCR_002798 | |
| Software, algorithm | u-track | *Jaqaman et al., 2008* https://doi.org/ 10.1038/nmeth.1237 | | |

*Continued on next page*

*Continued*

| Reagent type (species) or resource | Designation | Source or reference | Identifiers | Additional information |
|---|---|---|---|---|
| Software, algorithm | Matlab | The MathWorks, Inc | Version 9.6.0.1072779 (R2019a) | |
| Software, algorithm | BioRender | BioRender - biorender.com | | Used for *Figure 7a* |

## Expression constructs

Mouse Dishevelled2 (Dvl2; GenBank U24160.2; residues 2–736) and human Axin1 (NCBI NM_181050.3; our gene corresponds to residues 1–826 of NP_851393.1) genes were used to generate the constructs discussed below. Dvl2 DIX (12-92), Axin1 DAX (743-826), and DI-DAX (Dimerization/Inhibitory-DAX; 599–826) proteins were cloned in a modified pCDF-Duet-His$_6$-MBP-TEV vector with streptomycin resistance. DI-DAX Y760D construct was cloned in a modified pGEX-TEV vector with ampicillin resistance. Mammalian FLAG-Dvl2 constructs were cloned in the pCS2+ M2-FLAG vector with ampicillin resistance. Human tankyrase 2 SAM domain (residues 867–940) was cloned from pFLAG-TNKS-2 (NCBI NP_079511.1, 2–1166), which was a gift from Nai-Wen Chi (Addgene plasmid #34691). The *Archaeoglobus fulgidus* (NCBI WP_010878376.1 residues 1–77) Sm1 heptamerization sequence was synthesized as a codon-optimized mini-gene for human host cell expression (Integrated DNA Technologies, Coralville IA), which was used as a template for creating fusion proteins with Dvl2. Constructs were cloned using Gibson assembly (SGI DNA, LaJolla CA), overlap extension PCR, and restriction endonuclease digest/ligation (Fermentas, Waltham MA). Site-directed mutagenesis was primarily performed using PCR with primer pairs designed using Agilent Quick-Change; parental templates were digested with DpnI (New England BioLabs). Construct sequences are provided in *Supplementary file 1*.

## Expression and purification of His$_6$-MBP DIX, His$_6$-MBP-DAX, GST-DI-DAX Y760D, and MBP-DI-DAX

His$_6$-MBP-DIX variants, His$_6$-MBP-DAX variants, and GST or His$_6$-MBP-tagged DI-DAX variants were transformed into BL21(DE3) Codon-Plus RIL *Escherichia coli* (Agilent, Santa Clara CA). A single colony or a scraping from a glycerol stock was used to inoculate a starter culture. After 16 hr, the culture was expanded 1:100 up to 2L cultures in Luria broth and induced at A600 ~0.60 with 0.5 mM isopropyl β-D-thiogalactopyranoside (GoldBio, St. Louis MO) for 24 hr at 16°C. Cultures of MBP-tagged DIX or DAX in pCDF, and of GST-tagged DI-DAX in pGEX-TEV were grown in streptomycin (50 μg/mL) + chloramphenicol (34 μg/mL), and ampicillin (50 μg/mL), respectively. Cultures were harvested by centrifugation at 6000 x *g* for 15 min at 4°C. Harvested cell pellets were washed with ice cold 1x PBS and collected to be frozen with liquid nitrogen for storage at −80°C. Prior to lysis, for every 2L of culture, DNAse I (12 units; MilliporeSigma D5025) supplemented with 10 mM MgCl$_2$ was added to cells resuspended in lysis buffer containing 20 mM Hepes pH 8.0, 150 mM NaCl, 5% glycerol, and 4 mM dithiothreitol (GoldBio). Lysis was performed using an Emulsiflex homogenizer (Avestin, Toronto ON) with two passes through ice water-chilled tubing at a target pressure of 15,000–20,000 psi. Clarified lysate was collected after 50,000 x g centrifugation for 30 min and batch-bound for 1 hr under gentle rotation onto a pre-equilibrated agarose column at 10 mL bed resin per 1L culture. The loaded column was then washed with the same lysis buffer supplemented with 1 mM EDTA and 500 mM NaCl under gravity flow and re-equilibrated. Amylose agarose (New England BioLabs, Ipswich MA) and glutathione agarose resin (Pierce, Waltham MA) were used for purifying MBP-tagged and GST-tagged proteins, respectively.

All purification steps were carried out at 4°C or on ice. All buffers for protein purification were prepared with deionized MilliQ water, chilled ice-cold, and supplemented with a protease inhibitor mixture (1 mM phenylmethylsulfonyl fluoride, 21 μg/ml N-p-Tosyl-L-phenylalanine chloromethyl ketone, 42 μg/ml Nα-Tosyl-L-lysine chloromethyl ketone hydrochloride, 200 μM benzamidine, 150 nM aprotinin, 1 μM E-64, and 1 μM leupeptin). At each stage of purification, fractions were analyzed by stain-free SDS-PAGE (Biorad, Hercules CA). Coomassie blue and Sypro stains were used in cases where the protein construct was tryptophan-free. Reported protein concentrations were measured

by absorbance at 280 nm, BCA (ThermoFisher Scientific, Waltham MA), or Bradford (Biorad, Hercules CA) methods. All proteins were concentrated in Amicon Ultra centrifugal concentrators with regenerated cellulose membrane (MilliporeSigma).

### Production of soluble non-Filamentous DIX proteins

MBP or GST fusion proteins bound to 10 mL of amylose or glutathione agarose beads suspended in 10 mL lysis buffer supplemented with 1 mM EDTA were cleaved with ~0.6 mg of TEV protease under gentle rotation overnight at 4°C. The eluate was collected, and the beads were further washed with 3 mL of lysis buffer. The combined eluate was filtered through a 0.2 µm PES syringe filter and loaded onto a Superdex 75 26/600 size exclusion column, which was run in lysis buffer with 1 mM EDTA. DAX and DI-DAX were concentrated to ~300 µM and 100–200 µM, respectively.

### DIX filament production

Following column re-equilibration after a high-salt wash, MBP-DIX proteins were eluted with three column volumes of lysis buffer containing 10 mM D-(+)-maltose monohydrate (Sigma) applied to loose resin under gentle rotation for 30 min. After draining the first elution, one more column volume of elution buffer was mixed with the resin bed for 5 min to complete the elution. The combined eluate could be concentrated up to 15 mg/mL. Yields for soluble $His_6$-MBP-DIX range from 15 to 100 mg/L. For long-term storage at −80°C, the proteins were supplemented with 5% sucrose and subjected to flash-freezing with liquid nitrogen. For cleavage of $His_6$-MBP tag to trigger filament formation, 0.5 mg/mL TEV protease was mixed with His6-MBP-DIX at 1:35 v:v and incubated at 4°C overnight. Filaments were further purified using size-exclusion chromatography with a Superose 6 Increase 10/300 GL gel filtration column in lysis buffer containing 1 mM EDTA. For samples intended for electron microscopy, glycerol was omitted. Quantitative analysis of purified filaments by band intensity on stain-free SDS-PAGE gels indicates that the purified filament fractions contain ~10% uncleaved $His_6$-MBP-DIX. No electron density could be attributed to the MBP tag in the helical reconstruction.

### Negative stain electron microscopy

Fresh samples from peak fractions following size-exclusion chromatography (3 µL) were applied onto carbon-coated copper grids that had been glow discharged for 60 s. Grids were negatively stained with 5 µL of 0.75% w/v fresh uranyl formate for 30 s. Images were acquired using a 200-keV Tecnai F20 microscope (Thermo Fisher Scientific) equipped with a Gatan K2 Summit direct electron detector (Gatan, Pleasanton CA).

### Cryo-EM sample preparation and data acquisition

Filament samples were taken from the peak fraction of each Superose 6 size exclusion chromatography run. Sample concentration was optimized for best imaging density at 120 ng/µL (Dvl2 DIX wildtype and E22Q/E24Q) and at 70 ng/µL (Axin1 DAX NQ/NG).

Freshly prepared filament samples (3 µL) were applied to glow-discharged Lacey carbon film grids (LC200-Cu; Electron Microscopy Sciences, Hatfield PA). The grid was blotted for 2.0 s (DIX samples) or 1.5 s (DAX NQ/NG) using Whatman #1 filter paper (GE Healthcare) at 95% humidity, and then plunge-frozen into liquid ethane using a Leica EM GP (Leica Microsystems).

The wildtype Dvl2 DIX sample was imaged on a Titan Krios electron microscope (Thermo Fisher Scientific) operated at 300 keV. Movie-mode micrographs were recorded at a nominal magnification of 29,000x using a K2 Summit direct electron detector (Gatan, Pleasanton CA) in counting mode, with a size of 1 Å/pixel and a dose rate of ~9.8 $e^-/Å^2$/s. The total exposure time was 10 s, and each micrograph consisted of 50 frames. In total, 540 micrographs were manually collected with emphasis on medium-thickness ice. The images were motion corrected and dose weighted using MotionCor2 (*Li et al., 2013*) and Contrast Transfer Function (CTF) parameters were estimated by CTFFIND-4.1 (*Rohou and Grigorieff, 2015*).

The DIX QQ sample was imaged on a Tecnai F20 electron microscope. Movie-mode micrographs were recorded at a nominal magnification of 29,000 x using a K2 Summit direct electron detector in counted mode, with a pixel size of 1.286 Å/pixel on the specimen level and a dose rate of ~6.1$e^-/Å^2$/s. The total exposure time was 8 s, and each micrograph had 40 frames. Motion correction and dose

weighting were carried out as described for the WT DIX sample with the parameters given above. The.

The DAX NQ/NG sample was imaged on a Titan Krios electron microscope (Thermo Fisher Scientific) operated at 300 keV. Movie-mode micrographs were recorded at a nominal magnification of 22,500x using a K3 Summit direct electron detector (Gatan, Pleasanton CA) in super-resolution mode, with a pixel size of 1.096 Å/pixel and a dose rate of ~16 e$^-$/Å$^2$/s. The total exposure time was 2 s, and each micrograph had 40 frames. Motion correction and dose weighting were carried out as described for the WT DIX sample with the parameters given above.

## Cryo-EM data processing and molecular modeling

Processing was done in the SBGRID environment (*Morin et al., 2013*). Helical map reconstruction was performed with RELION 3.0.8 (*He and Scheres, 2017*; *Scheres, 2012*; *Zivanov et al., 2018*). Semi-automated particle picking on micrographs with defocus between 0.5 and 2.0 μm and estimated maximum resolutions < 4 Å yielded 437,872 particles. Successive rounds of reference-free 2D and 3D classification enabled selection of 110,105 particles from classes with well-defined features. The selected particles were subjected to 3D auto-refinement, Bayesian polishing and post-processing (map sharpening) to produce a final map with a global resolution estimate of 3.6 Å by the 0.143 Fourier shell correlation criterion (*Rosenthal and Henderson, 2003*; *Figure 2—figure supplement 1d*). The mask used for 3D refinement and sharpening was generated by extending a map 15 pixels from a preliminary model of all twelve subunits in Chimera (*Pettersen et al., 2004*). Using the 'mask create' job type in RELION, the map was then lowpass filtered to 15 Å and the binarization threshold set to 0.06 before application of a binary edge of 1 pixel and a soft edge of 4 pixels. The final map was symmetrized according to the refined helical rise and twist after post-processing using the –`impose` option in the RELION_helix_toolbox. Local resolution (*Figure 2—figure supplement 1e*) was calculated from the unsharpened map in RELION.

We used the crystal structure of Dvl2 DIX Y27W/C80S (PDB 6IW3, with the Trp and Ser changed back to Tyr and Cys) to initially fit the cryo-EM map by rigid body refinement, and then used real space refinement in Phenix (*Afonine et al., 2018*) and manual model building in Coot (*Emsley et al., 2010*) to obtain the final structure (*Table 1*; *Figure 2—figure supplement 1f*). Because the model was refined against the full map, its coordinates were first randomized with a mean error value of 0.2 Å. It was then refined against one half map, and Mtriage was then used to calculate the FSC between the model and each half map, as well as the full map (*Figure 2—figure supplement 1g*). Figures were generated with PmMOL (Schrödinger, LLC) and Chimera (*Pettersen et al., 2004*).

Coordinates of the Dvl2 DIX filament have been deposited in the PDB, code 6VCC, and the cryo-EM map in the EMDB, code EMD-21148.

## DIX filament sedimentation assays

Purified Dvl2 DIX filament samples with or without Axin DAX were incubated at room temperature for 30 min, then centrifuged at 386,000 x *g* for 7 min at 4°C. After removal of the supernatant, the pellet was resuspended by vortexing in 1X SDS-PAGE sample buffer for 30 s. Pellet and supernatant fractions were then run on a 15% Tris-glycine-SDS gel containing 6M urea. The gel was stained with Coomassie blue and imaged on a LiCOR scanner (LICOR, Inc, Lincoln, NE), and band intensities determined in FIJI (*Schindelin et al., 2012*). Data from biological and technical replicates were analyzed in GraphPad PRISM 8. Error bars represent the standard error of the mean.

For filament formation assays, MBP-DIX at the indicated concentrations was incubated with 27 ng/μL TEV protease overnight at 4°C, and the samples were then pelleted as described above. Supernatant and pellet samples were run on a NuPage 4–12% Bis-Tris gel (ThermoFisher Scientific), which was visualized with Sypro Ruby stain (ThermoFisher Scientific, Waltham, MA) using a BioRad Gel Doc EZ Gel Imager (BioRad, Hercules, CA). Band intensities were quantified and plotted as described above.

## Sizing of DAX and DIX domain variants on Superdex 75

For the data in *Figure 3—figure supplement 1*, 500 μL samples were injected onto a Superdex 75 10/300 column that was pre-equilibrated with 25 mM HEPES, pH 8.0, 150 mM NaCl, 2 mM DTT, 1

mM EDTA, and 5% glycerol. The samples were run at 0.5 mL/min. Runs showing multiple peaks were analyzed by SDS-PAGE, followed by Coomassie staining.

## Multiangle light scattering

The molecular weights of Dvl2 DIX, Axin1 DAX or DI-DAX samples (100 μL) were determined by size exclusion chromatography coupled with inline multi-angle light scattering (MALS) using a Superose 6 10/300 (for Dvl2 DIX) or Superdex 200 10/300 GL (for DAX and DI-DAX) column attached to a UV detector, followed by a DAWN Heleos-II and an Optilab T-rEX refractive index (RI) detector (Wyatt Technology, Santa Barbara CA). The system was equilibrated with 25 mM HEPES, pH 8.0, 150 mM NaCl, 2 mM DTT, 1 mM EDTA, and 0.01% $NaN_3$ at 25°C. 5% glycerol was included in experiments with Axin1 DAX, and 400 mM NaCl was used for DI-DAX. Detectors were calibrated by measuring the signal of monomeric bovine serum albumin at ~70 kDa. The absolute mass over the course of the run was determined with ASTRA 6 software (Wyatt Technology) using the signals from the MALS and the RI detectors. Concentrations at the detector were determined from the maximum dRI of the peak.

## Native protein gel electrophoresis of DIX-DAX mixed samples

Purified DIX filament and DAX samples were mixed at the indicated concentrations and incubated for 30 min at room temperature. To each 15 μL mix, 5x native loading buffer (250 mM Tris-HCl, pH 6.8, 50% glycerol, no loading dye) was added. Samples (14 μL) were loaded into lanes where indicated of an Any kD Mini-PROTEAN TGX 15-well gel (#4569036, Biorad, Hercules CA). Electrophoresis was carried out in native Tris-glycine buffer (25 mM Tris, 190 mM glycine, pH ~8.3) at 4°C at 25V for 12 hr followed by an additional 2 hr at 100V. DIX filaments labeled with AlexaFluor-488 C2 maleimide (with some residual dye left over from using a 2 mL 7,000 MWCO Zeba spin desalting column) were visualized using the SYBR Green setting of a Biorad Gel Doc EZ instrument with a 10 s exposure. To confirm specific detection of DIX fluorescence, DAX samples were run on SDS-PAGE and were visualized using the same acquisition and image contrast settings. Both native- and SDS-PAGE gels were stained with Coomassie Blue for total protein.

## Wnt signaling assays

The activities of full-length Dvl constructs were tested using a Wnt-responsive TOPFlash luciferase reporter assay by transient expression in Dvl TKO HEK293T cells (generously provided by Stephane Angers, University of Toronto). Cells were seeded in a white opaque CulturPlate-96 (Perkin Elmer, Waltham MA) in DMEM containing 10% (v/v) FBS. After 6 hr, cells were transiently co-transfected with a SuperTOPFlash plasmid, as well as the indicated Dvl mutants, using Lipofectamine 2000 (Life Technologies) according to the manufacturer's protocol. The media were replaced with either conditioned media produced from mouse L cells (control) or from L cells expressing Wnt3a at 18 hr post-transfection, supplemented with 20 mM HEPES pH 8.0. After another 18 hr, Luciferase reporter luminescence was measured in a Synergy two plate reader (BioTek, Winooski VT). Activity assays for each Dvl construct were carried out at least three times, each with triplicate technical replicates. To report levels of Wnt-dependent Dvl2 signaling, each measurement in relative luciferase units was normalized to the level of wild type Dvl2 stimulated with Wnt3a conditioned L cell media. Dvl constructs were expressed at near endogenous levels, determined by using wildtype HEK293T cell lysate as a reference, blotted with anti-Dvl C terminus antibody (#3216, Cell Signaling, Danvers MA) and measuring the intensity of the Dvl band determined using a LICOR scanner (LICOR, Inc, Lincoln, NE) (see *Figure 3—figure supplement 1*). Data from biological and technical replicates were analyzed in GraphPad PRISM 8. Error bars represent the standard error of the mean.

## CRISPR/Cas9 mediated endogenous tagging of Dvl2 and Axin1

Endogenously tagged HEK293T cells were generated essentially as described (*Sakuma et al., 2015*). In brief, DVL2 and subsequently Axin1 were C-terminally tagged at their endogenous loci using CRISPR/Cas9 gene editing and micro-mediated homologous end joining (MMEJ) to introduce GFP-T2A-PURO and RFP-T2A-HYGRO, respectively. The CRISPR guide design tool crispr.mit.edu was used to design targeting guides (DVL2 – CAATCCCAGCGAGTTCTTTG; Axin1 – CATCGGCAAAG TGGAGAAGG), which were cloned into pX330A-1 × 2 (Addgene #58766) and combined with

pX330A-2-PITCh (Addgene #63670) by golden gate cloning. To generate the DVL2 repair construct, DVL2 micro-homology arms were cloned into pCRIS-PITChv2 (Addgene #63672) either side of GFP-T2A-PURO. To generate the Axin1 repair construct Axin1 micro-homology arms and T2A sequences were incorporated into overlapping primers used to amplify RFP from ds-RED-N1 and HYGRO from pcDNA5/FRT/TO and assembled into pCRIS-PITChv2 by Gibson Assembly.

HEK293T cells (50–60% confluence) were transfected with pX330A/PITCh and pCRIS-PITChv2 (2:1) using PEI, the media was changed the next day, and selection was started 72 hr post-transfection (1 μg/ml puromycin or 100 μg/ml hygromycin) for 7 days. After recovery, GFP/RFP positive cells were sorted into individual clones by flow cytometry. Clones were analyzed by local genomic DNA sequencing and expression was confirmed by western blot analysis using anti-DVL2 (CST #3216) and anti-Axin1 (CST #C76H11) antibodies.

## TIRF microscope

Total internal reflection fluorescence (TIRF) microscopy was done with an inverted microscope (Nikon TiE) using an Apo TIRF x100 oil, NA 1.49, objective lens (Nikon) and was controlled through Micromanager (*Edelstein et al., 2014*). Experiments were conducted at room temperature for the in vitro filament assays and using an objective heater (Bioptechs) set at 37°C for the live cell experiments. The microscope was equipped with a Perfect Focus System. A red laser (635 nm, Blue Sky Research) and a blue laser (473 nm Obis, Coherent) were used for Alexa Fluor 647 and GFP excitation respectively. Emitted light went through a quad-edge laser-flat dichroic with center/bandwidths of 405 nm/60 nm, 488 nm/100 nm, 532 nm/100 nm, and S5 635 nm/100 nm from Semrock (Di01-R405/488/532/635−25 × 36) and a corresponding quad-pass filter with center/bandwidths of 446 nm/37 nm, 510 nm/20 nm, 581 nm/70 nm, 703 nm/88 nm bandpass filter (FF01- 446/510/581/703–25). An additional filter cube (679 nm/41 nm, 700 nm/75 nm for Alexa Fluor 647 imaging; 470 nm/40 nm, 495 nm LP, 525 nm/50 nm for GFP imaging) was included before the camera in the light path. All analyzed images were taken at 100 ms exposure time using the full chip of a Hamamatsu Orca Flash 4.0 camera (chip size 2048 × 2048 pixels). The average background in different ~20 μm boxes varied ~10% over the field of view.

## TIRF imaging of purified DIX filament severing by DAX

For fluorescent labeling of DIX at Cys80 (the only Cys in the MBP-DIX construct), 500 μL of His$_6$-MBP-DIX at ~10 mg/mL was exchanged into buffer containing 50 mM HEPES, pH8.0, 150 mM NaCl, 1 mM EDTA, and 4 mM TCEP using a 2 mL 7,000 MWCO Zeba spin desalting column (ThermoFisher Scientific, Waltham MA). The protein was incubated with 40 μL of 10 mM AlexaFluor-647 C$_2$ maleimide (ThermoFisher Scientific, Waltham MA) (stock dissolved in DMSO) overnight at 4°C. The reaction was quenched by the addition of DTT to a final concentration of 10 mM, and the protein was cleaved for one hour at room temperature by adding 0.5 mg/mL TEV protease at 1:35 v/v. Labeled filaments were further purified using size-exclusion chromatography with a Superose 6 Increase 10/300 GL gel filtration column in lysis buffer containing 1 mM EDTA. Functionality of labeled DIX filaments was verified by negative stain EM, resistance to serial dilution and DAX solubilization assays (data not shown). Dye coupling efficiency to DIX Cys80 was >50% based on the relative extinction coefficients of the protein and dye.

A total of four replicate imaging experiments were performed, two per day, using two separate preparations of DIX filaments and a single DAX preparation. For each experiment, Alexa 647 labeled DIX filaments were incubated with (experimental) or without (control) DAX at room temperature for ~1 hr (sample 1) or ~1.30–2 hr (sample 2) before the start of imaging. Samples were placed in 35 mm glass bottom dishes with 14 mm micro-wells (#1.5 cover glass) from Cellvis. Each micro-well was prepared by first washing with DIX gel filtration buffer ('DIX buffer') at least three times, then filled with 100 μl DIX buffer. A 1 μl sample containing 10 μM DIX with or without 50 μM DAX was diluted 1:100 with DIX buffer and immediately added to the micro-well for a final dilution of 1:200. After a 30 s incubation, the micro-well was washed at least six times by two-fold dilution with buffer to wash away free proteins in solution and left in 100 μl buffer before placing the micro-well on the microscope. Imaging started 1–5 min after washing and was completed within 1–5 min. Most of the images were taken by walking the stage diagonally across from one corner of the coverslip to the other, avoiding regions that had been previously bleached. For replicate 1, we imaged the control

before the experimental sample and did the reverse for replicate 2. Imaging was performed at laser powers out of the objective of 0.44 mW (day 1) and 5.1 mW (day 2). The results from the two days were similar, so we concluded the data are not strongly dependent on laser power over this range.

Images were processed in FIJI (*Schindelin et al., 2012*). Each pair within a replicate had 18 to 63 image frames. The mean gray values MGVs of each frame were recorded. We compared the distribution of MGVs using the Mann–Whitney U test, which gave $p<0.05$ for each replicate. A bootstrapping test was run to reject the null hypothesis that the means were not different. We resampled data points for each condition within a pair and recorded the difference of means 5000 times. The distribution for each replicate indicated a positive difference of means (experimental case with higher background) for the middle 95% of each replicate.

We recorded the ratio of the means for each replicate. We used bootstrapping (resampling 5000 times) to get a distribution for the means that was fit to a Gaussian. The standard deviation of each mean was propagated to produce error values for each ratio of means. The 4 values of the ratio of means were then averaged to get the mean ratio of means with errors from the previous step propagated.

## Immunofluorescent staining for β-catenin accumulation

Dvl2-GFP/dsRed-Axin1 HEK293 cells were seeded at ~60% confluence in Lab-Tek chamber slides (Nunc 177380) that were pre-coated with 0.1 mg/mL poly-D-lysine (Sigma P6407). The following day, the cells were washed with Hanks' balanced salt solution (HBSS) (Gibco 24020117) and then treated at 37°C with L cell control or Wnt3A-containing conditioned media for 2 hr. At the end of treatment, cells at room temperature were fixed for 10 min with 2% formaldehyde (Electron Microscopy Sciences 15710), and permeabilized for one hour with 0.1% Triton X-100 (Biorad 1610407) supplemented with 1% BSA (Sigma). Cells were then stained overnight at 4°C with AlexaFluor 647-conjugated anti-β-catenin antibody L54E2 at 1:300 dilution supplemented with 3% BSA (2677S, Cell Signaling). Between and after each of these steps, cells were washed three times with HBSS, which was also used to make up each reagent. Sample mounting was achieved using ProLong glass anti-fade mountant with NucBlue (Invitrogen P36983), and each chamber was sealed with a glass coverslip. Cell epifluorescence was imaged at 60x and AlexaFluor647 was detected on the Cy5 channel. For every condition, 5–7 fields of view, each with ~100 cells, were acquired under the same exposure. Mean gray values (MGVs) at the center of each frame were measured and tallied per condition for comparison as shown in *Figure 7—figure supplement 1d*.

The collection of the MGVs in the center (1000-by-1000 pixels) of each frame in the ±Wnt conditions were significantly different as assessed by the Mann-Whitney U test ($p<0.05$). A bootstrap test was run to reject the null hypothesis that the means were not different where the collection of points were resampled at the sample size and difference of means calculated for 500 runs. The middle 95% of the resulting mean difference distribution was positive indicating a significant rise in β-catenin levels.

## Cell culture for live cell imaging

For the short-term epifluorescent live cell imaging and the apical single molecule localization microscopy assay, HEK293T Dvl2-GFP/Axin1-RFP cells were passaged no more than two times after expansion, incubated in sterile-filtered DMEM media containing phenol red (15 mg/L), high glucose (4500 mg/L), L-glutamine (584 mg/L) (Thermo Fisher Scientific, 11965092), with added 10% FBS, Sodium Pyruvate (110 mg/L), and 1% penicillin–streptomycin (Thermo Fisher Scientific, 15140122). Cells were passaged through splitting by cell dissociation using 0.25% Trypsin-EDTA (Life Technologies, 25200056) and subculturing. L control conditioned media, and L Wnt3A-conditioned media are described above in the Wnt signaling assay section. For the long-term epifluorescent live cell imaging, the cells were passaged 3 to 4 times in DMEM based media with same supplements as above but with HEPES based $CO_2$-independent buffering (5958 mg/L) (Thermo Fisher Scientific, 21063045) instead of phenol red.

## Epifluorescence microscope

An inverted Nikon Ti-E microscope was controlled using Micromanager (*Edelstein et al., 2014*). The microscope was connected to a pre-equilibrated heating chamber set to 37°C and equipped with a

Perfect Focus System, a Heliophor light engine (89 North) and an Andor sCMOS Neo camera. We utilized two objective lenses: a CFI Plan Apo Lambda x40 0.95 air objective lens; and a CFI Plan Apo Lambda x60 1.40 air objective lens. Acquisitions were performed on GFP and Cy5 channels as appropriate.

## Live cell epifluorescence imaging

For short term imaging, we seeded cells overnight onto two wells with passaging media (DMEM with 10% FBS) and two wells with L control conditioned media in a 24-well glass bottom plate (FisherScientific NC0397150). Before imaging, we changed one well each from the passaging or L control media into L Wnt3A-conditioned media. Multiple regions of the Wnt-stimulated cells were imaged at 60x magnification at 10–20 min and 40–50 min. We also imaged multiple regions of the control cells at the start and end of the experiment. All fluorescence images used for manual counting were acquired with 300 ms exposures.

For long-term epifluorescence imaging (*Video 1*), cells were seeded overnight in a 24-well glass bottom plate (FisherScientific NC0397150). We recorded coordinates of multiple regions within and across wells and sequentially acquired images from all the regions every ~3 min for ~16 hr at 40x magnification.

## Apical TIRF live cell imaging for Dvl2-GFP tracking and intensity analysis

All experiments were conducted using an objective heater (Bioptechs) equilibrated at 37°C and all media used at all stages were also equilibrated at 37°C. Two sets of experiments were run. In the first set, cells were imaged before and after stimulation with Wnt3A conditioned media, whereas the second set were replicates using only Wnt3A stimulated cells. In the first set, cells were seeded overnight in passaging media on the outer facing part of the filter of transwell inserts (12 mm Transwell, 0.4 µm Pore Polyester Membrane Insert, Corning, 3460). The transwells were kept inverted in 6-well flat bottom plates to ensure cell settling (Corning CoStar, 3506) and incubated in ~200 µL media for 1–2 days. The empty wells were filled with media for humidity. Whenever needed, the 6-well plate lid was placed and removed with care to avoid perturbing the media sandwiched between the lid and the transwell tops. Before imaging, a 35 mm glass bottom dish with 20 mm micro-well (#1.5 cover glass) (Cellvis, D35-20-1.5-N) was equilibrated at 37°C with 1.5 ml passaging media. The transwell in the experiment was inverted gently using tweezers into a 12-well flat bottom plate (Corning CoStar, 3512) that was used as a stable carrier for transfer to imaging room, with cells apical side down inside ~0.5 ml passaging media. Immediately after inversion, 1 ml passaging media was added to the inside of the transwell filters. A custom designed 3D printed holder allowed stable placement of the glass bottom dish on the microscope. After removal of the lids of the plate and the glass bottom dish, the transwell insert was quickly and gently transferred with sterile tweezers onto the micro-well, with cells still apical side down. The perfect focus system would not always work as desired with the transwell insert above the glass, and it was sometimes necessary to find the correct plane before placement of the insert onto the micro-well. Imaging commenced immediately following placement of the transwell insert. After imaging a number of regions, the transwell insert was gently lifted, the micro-well was spiked with 0.5 ml L Wnt3A-conditioned media equilibrated at 37°C, and the transwell insert was gently positioned back. Imaging continued for up to ~70 min.

In each of the five experimental pools, 4–11 analyzed fields of view with multiple cells were imaged, each for about fifty to a few hundred frames. We used multiple configurations: a) 2–3 different TIRF angles, b) focus centered as close to the coverslip as possible or slightly above (~200 nm), c) at two different blue laser powers (3 mW and ~5 mW out of the objective). Regions were not necessarily illuminated by the strongest part of the TIRF field due to separation from the cover glass, yet we were still able to image at high signal-to-noise ratio.

During this first set of experiments, we noticed that lifting the PET filter and flowing in Wnt3A-conditioned media into the microwell occasionally caused cell blebbing. We also sought to investigate the case where both sides of the cells were exposed to Wnt3A-conditioned media. Thus, the protocol was slightly altered for the second set of experiments. We seeded the cells on transwell inserts into media conditioned by control L cells instead of the passaging media. Before imaging, the glass bottom dish was filled with 1.5 ml L cell Wnt3A-conditioned media, and the inside of the

transwell was also filled in with 1 ml L cell Wnt3A-conditioned media while in the 12-well carrier plate, with the time of first Wnt3A exposure noted. Within ~2 min, the transwell insert was gently placed on to the micro-well (filled with L cell Wnt3A-conditioned media), and then imaged.

We pooled data from videos according to the following criteria: The first 40 frames (100 ms exposure per frame) were pooled when belonging to movies recorded the same day, with the same laser power and approximate TIRF angle and condition (with or without L cell Wnt3A-conditioned media treatment). This produced 2 pools of data without and three pools with L cell Wnt3A-conditioned media treatment. When analyzing data from L cell Wnt3A-conditioned media-treated cells, we pooled all videos starting at 9 min after the addition of L cell Wnt3A-conditioned media.

The 40 frames for each region in a given pool were processed in Fiji (*Schindelin et al., 2012*) to apply a rolling ball background filter (five pixel/325 nm diameter). We cropped out large empty regions where there were no cells in the TIRF field. We then used u-track (*Jaqaman et al., 2008*) in Matlab in order to extract single particle trajectories. We optimized the detection and tracking parameters and confirmed reasonable tracking by eye. Specifically, for detection, we chose the single particle detection with Gaussian mixture model fitting option with a Gaussian standard deviation one pixel. We selected rolling window time-averaging of window size five and an alpha-value of 0.05 for comparison with local background. For tracking, the maximum gap to close was set to four frames.

The output from u-track was analyzed using custom Matlab code. For each pool, we extracted spot trajectories 10 frames or longer which was helpful in getting rid of the background from the media while keeping most of the true positives. We recorded the intensity at the starting frame of those trajectories. Theoretically, for the subpopulation of spots with $n$ fluorophores, the background subtracted intensity distribution will be a single Gaussian, peaking roughly at $n$ times the mean intensity of a single fluorophore. Thus, the intensity distribution from the ensemble of spots was fit to a Gaussian mixture model for one through N components using the built-in 'fitgmdist' function with 100 replicates and a maximum of 500 iterations, and the number of components minimizing Bayesian Information Criterion (BIC) or Akaike Information Criterion (AIC) was selected. N, the largest number of possible components tested for a given pool, was always set larger than the number of components eventually found to provide the best fit (all cases with BIC and some with AIC), or alternatively one less than the number of components that frequently resulted in ill-conditioned covariances in every replicate of a run (as sometimes occurred with AIC). While the criteria resulted in fits different in the number of mixture components and their means, they gave similar final normalized distributions (see below), with 90% of the spot intensities being within 10x of a single GFP intensity in both cases. The plot in *Figure 7c* utilizes BIC for model selection.

Due to the observed cell-to-cell heterogeneity in the HEK293T Dvl2-GFP/Axin1-RFP cell line, the limitations of the GFP tag, and the possibility of the Dvl2 complex size having a highly skewed distribution, we did not focus on the proportions of different mixture components which would map to different proportions of complex sizes. Instead, we treated the difference of the two smallest mixture component means from the fit as the single GFP mean intensity value, subtracted the background value this implied from its difference from the minimum mixture component mean, and then normalized the intensity axis accordingly (*Figure 7c*). The single GFP value for each pool was always smaller (within ~2 fold in the BIC case) than the smallest mixture component mean, which we attribute to imperfect background subtraction, and bleaching within camera exposures. We note that we expect a typical maturation efficiency of 90% for eGFP (*Cormack et al., 1996*) to affect our estimations for numbers of molecules by ~10%.

Fluorescence images and FIJI macros are available upon request. Matlab code is provided in Source code files 1 and 2.

## Acknowledgements

We thank Stephane Angers for providing the HEK293T Dvl TKO cells; David Bushnell, Elizabeth Montabana, Qianhui Qu, Alpay Seven and Min Su for assistance and advice in the EM analysis; Sabine Pokutta for advice on the sedimentation assays; and Jake Mahoney for discussions and comments on the manuscript.

## Additional information

### Competing interests

William I Weis: Reviewing editor, *eLife*. The other authors declare that no competing interests exist.

### Funding

| Funder | Grant reference number | Author |
|---|---|---|
| National Institute of General Medical Sciences | GM119156 | William I Weis |
| National Institute of General Medical Sciences | GM130332 | Alexander R Dunn |
| National Institute of General Medical Sciences | T32 GM007276 | Michael D Enos |
| Pew Charitable Trusts | Pew Scholars Innovation Award 00031375 | Georgios Skiniotis William I Weis |
| Stanford Bio-X Graduate Fellowship | Graduate fellowship | Elgin Korkmazhan |
| Fritz Thyssen Foundation | Postdoctoral Fellowship | Stefan Muennich |
| HHMI Faculty Scholar | | Alexander R Dunn |
| Medical Research Council | MC_U105192713 | Mariann Bienz |
| Cancer Research UK | C7379/A15291 | Mariann Bienz |

The funders had no role in study design, data collection and interpretation, or the decision to submit the work for publication.

### Author contributions

Wei Kan, Michael D Enos, Conceptualization, Formal analysis, Investigation, Methodology, Writing - original draft, Writing - review and editing; Elgin Korkmazhan, Conceptualization, Software, Formal analysis, Investigation, Methodology, Writing - original draft, Writing - review and editing; Stefan Muennich, Formal analysis, Investigation, Methodology; Dong-Hua Chen, Methodology; Melissa V Gammons, Resources, Methodology; Mansi Vasishtha, Investigation; Mariann Bienz, Resources, Writing - review and editing; Alexander R Dunn, Supervision, Investigation, Methodology, Writing - review and editing; Georgios Skiniotis, Supervision, Methodology, Writing - review and editing; William I Weis, Conceptualization, Formal analysis, Supervision, Funding acquisition, Writing - original draft, Project administration, Writing - review and editing

### Author ORCIDs

Wei Kan https://orcid.org/0000-0002-6830-6714
Elgin Korkmazhan http://orcid.org/0000-0002-6872-9952
Stefan Muennich http://orcid.org/0000-0003-1355-737X
Mariann Bienz http://orcid.org/0000-0002-7170-8706
Alexander R Dunn http://orcid.org/0000-0001-6096-4600
William I Weis https://orcid.org/0000-0002-5583-6150

### Decision letter and Author response

Decision letter https://doi.org/10.7554/eLife.55015.sa1
Author response https://doi.org/10.7554/eLife.55015.sa2

## Additional files

### Supplementary files

• Source code 1. Single molecule analysis code to analyze pools individually in the apical TIRF live cell Dvl2-GFP imaging/intensity experiments.

- Source code 2. Single molecule analysis code to combine different pools and plot data for the apical TIRF live cell Dvl2-GFP imaging/intensity experiments.

- Supplementary file 1. Plasmid sequences. Nucleotide sequences for constructs discussed in this paper. There are two classes of plasmids: those used to make recombinant protein in bacteria in the pCDF MBP-TEV or pGEX-TEV vector backbone, and those used to express full-length Dvl proteins in cells for signaling assays, which are in the pCS2+ vector backbone under the control of the SP6 promoter. The full sequence of each vector backbone is provided, including appropriate sequencing primers for each vector backbone. For analogous constructs between mammalian and bacterial expression, please refer to the bacterial expression constructs for the exact nucleotide sequence. Yellow highlights indicate either the mutated residue or extra sequences inserted inside the native open reading frame.

- Transparent reporting form

## Data availability

Coordinates of the Dvl2 DIX filament have been deposited in the PDB, code 6VCC, and the cryo-EM map in the EMDB, code EMD-21148.

The following datasets were generated:

| Author(s) | Year | Dataset title | Dataset URL | Database and Identifier |
|---|---|---|---|---|
| Kan W, Enos M, Muennich S, Skiniotis G, Weis WI | 2020 | Dvl2 DIX filament coordinates | https://www.rcsb.org/structure/6VCC | RCSB Protein Data Bank, 6VCC |
| Kan W, Enos M, Muennich S, Skiniotis G, Weis WI | 2020 | Dvl2 DIX filament cryo-EM map | https://www.ebi.ac.uk/pdbe/entry/emdb/EMD-21148 | Electron Microscopy Data Bank, EMD-21148 |

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
