## [Decision Letter]

**Acceptance summary:**

Thanks for the efforts to address the few minor points regarding your revised manuscript. Both reviewers feel that it is a very strong piece of work and will receive well-deserved attention from readers in the Wnt signaling field and in the field of protein module-mediated molecular assembly formation.

**Decision letter after peer review:**

Thank you for submitting your article "Limited Dishevelled/Axin oligomerization determines efficiency of Wnt/b-catenin signal transduction" for consideration by *eLife*. Your article has been reviewed by two peer reviewers, including Mingjie Zhang as the Reviewing Editor and Reviewer #1, and the evaluation has been overseen by Cynthia Wolberger as the Senior Editor. The following individual involved in review of your submission has agreed to reveal their identity: Mark Peifer (Reviewer #2).

The reviewers have discussed the reviews with one another and the Reviewing Editor has drafted this decision to help you prepare a revised submission.

As you can see, both reviewers are quite enthusiastic about the work described in the manuscript. The comments from the two are quite detailed, and these comments should help to improve the quality of the manuscript in the revised form. Since the comments from the two reviewers are complementary, we decided not to merge the comments in this case. Please revise your manuscript to address these comments.

Reviewer #1:

In this study, Kan et al., characterized the structural properties of the DIX domains from Dvl2 and Axin. Using cryo-EM, the authors determined the oligomer structure of Dvl2 DIX and revealed that Dvl2 DIX form a two-stranded, antiparallel oligomer. The cryo-EM structure also reveals the molecular basis governing the inter-strand interactions that stabilize the double strand conformation of Dvl2 DIX. The authors went on to discover that the Axin DIX domain (DAX) can break/sever Dvl2 DIX oligomer via capping the ends of the Dvl2 DIX oligomer, which is a rather unexpected finding. The authors showed that formation of small Axin/Dvl2 DIX heteroligomer is critical for the transcriptional activity of Dvl2 in a cell-based assay. Finally, the authors provided evidence showing that in living cells the endogenous Dvl2 only form small oligomers (with 10 or less protomers) regardless of Wnt signal stimulation. With these findings, the authors proposed a new model showing how DIX domain-mediated Dvl/Axin oligomerization can transduce Wnt signaling.

The results presented in the current study are potentially interesting and valuable for understanding Wnt signaling. However, the manuscript contains a number of issues as detailed below. These issues dampen the rigorous of the conclusions drawn, and thus will need to be addressed.

Major issues:

1) Figure 1B and C: the pellet fraction of Dvl2 DIX was observed to increase as a function of protein concentration decrease at the protein concentrations below 6.25 µM. Why could this occur? Related to the above point, the dilution experiment shown in Figure 1D is also strange. At the intermediate dilution folds (e.g. around the final concentration ~3.7 µM), one can see obvious depolymerizations of Dvl2 DIX oligomer. Such depolymerization seems to be less obvious at both high and low final concentrations of the dilution experiment.

2) Figure 2C: The labeling of the figure needs to be improved. It will be helpful for recognizing the head and tail of each subunit in Figure 2C if the secondary structure unit of one protomer (e.g. the brown one at the upper left corner) can be labeled. Similarly, it would help if the strand names containing Y27 and K68 in Figure 2D are labeled. Similarly, secondary structures in Figures 3B and D should be labeled for easy reading.

3) Figure 3 and 4: The SEC analysis in the two figures are complicated by the cleaved MBP in the mixtures. For example, the authors state that the Y27D mutant of Dvl2 DIX is a monomer, but the elution peak overlaps with the MBP tag, which is considerably larger than the monomeric Dvl2 DIX. The authors should remove MBP from the cleaved protein mixtures and re-analyze the elution behaviors of the mutants.

4) Figure 3: The G65D mutant is interesting as the mutation seems to disrupt both inter- and intra-strand protomer DIX interaction. This brings up two issues: the mutation cannot really be used to support the inter-strand interaction observed in the cryo-EM structure; and the mutation might have a large overall folding perturbation of the Dvl2 DIX protomer. The authors should check the folding of the G65D mutant. Additionally, the G65D mutant, though not capable of forming DIX oligomers, showed a comparable level of transcriptional activity to that of the M60A mutant (Figure 3C). This will need to be explained.

5) Figure 4 and paragraph describing data in Figure 4 (subsection “Axin DAX oligomerizes through a head-to-tail interface but does not form large polymers”) is confusing. If the pink elution profile was obtained for DAX at 180 µM, why the highest concentration tested was only 24 µM? Even for the lowest concentration (0.7 µM) analyzed, the peak would likely be an ensemble average of different oligomerization states of DAX instead of simple monomer/dimer equilibrium. If this is true, estimation of apparent Kd of 0.7 μm would not be appropriate. Consequently, a number of places in the manuscript concerning the affinity of DAX association will need to be re-written.

6) Figure 3—figure supplement 1C and D (related to the question concerning Figures 3 and 4): It appears that each cleaved Dvl2 DIX mutant was eluted as an oligomer and a smaller multimer on SEC column. This is a bit odd. Do these two populations exchange? Or each population is trapped in their kinetically stable form?

7) Figure 5A: why ~40% of Dvl2 DIX was recovered in the pellet/filament form even in the presence of very high concentration of DAX? Related to this question, it would be informative to perform an SEC analysis of Dvl2 DIX with increasing concentrations of DAX (or Y760D DAX) to check the hetero-DIX oligomer formation status.

8) The data for the "Avidity for Axin enhances signaling" and Figure 4D is not convincing and interpretation is tentative. The data seem to indicate that as long as Dvl2 and Axin can form heteromer, the Dvl2 activity can be realized. Additionally, the enhanced heteromer formation between Dvl2 and Axin seems to promote Dvl2 transcription activity. The use of TNKS2-SAM potentially brings in another indirect binder to the system. The interpretation presented in the current manuscript based on the data at the present form is one of many possibilities. It would be nice to use a pair of small heterodimer forming protein domains to replace Dvl2 and Axin DIX domains, respectively, and then to repeat the transcription assay. This is a demanding experiment, the authors should at least consider similar line of assay or any assay that would provide a more definitive result to support their model.

Reviewer #2:

Wnt signaling is one of the handful of signaling pathways that play key roles in embryonic development, tissue homeostasis and which go wrong in cancer. While the field created an outline of the signaling pathway two decades ago, many unanswered questions remain. The key regulated effector is the protein β-catenin, which, in the absence of Wnt ligands, is targeted for phosphorylation and ultimate proteolytic destruction by the multiprotein destruction complex. One key open question involves the molecular mechanisms by which Wnt ligands turn down destruction complex activity. This requires the cytosolic effector Dvl, but its mechanism of action remains unclear. One critical issue is whether and how Dsh:Dsh, Axin:Axin, and Dsh:Axin multimerization regulate the balance of destruction complex activity. Here Weis and colleagues present the results of a multidisciplinary tour-de-force investigation, combining cryo-EM, powerful analytical biochemistry, and in vivo functional and imaging observations. They present some very intriguing new insights, revealing the polymerization of Dsh's DIX domain into anti-parallel double helical filaments, careful investigation of the parameters of polymerization in the presence and absence of Axin, clever structure-based mutational analyses that reveal important features of homo- and hetero-polymerization, combined with in vivo assays of function. These data provide a firm biochemical basis and important molecular constraints for previous speculative models for Dsh/Axin interactions, significantly advancing our understanding of this key signaling pathway. In general, their data are lovely, well controlled and quantitatively analyzed, and their conclusions well founded. As I lay out below, I think at times they are a bit to enamored with their in vitro data, making broad conclusions about events in vivo that could be presented in a more nuanced way by considering other data in the field. With these and some other minor modifications, I think this work will be of exceptional interest to cell, developmental and cancer biologists.

Data issues

Figure 1. This was lovely work but I found one thing puzzling. Why was the pelleting profile of Dsh biphasic, with significant pelleting at low concentrations (<1µm)?

Figures 2-4. As I will note below in comments on the model, the authors come down very firmly on the idea that the "active" Dsh is in short polymers of less than 10 monomers. Their structural studies are stunning, and the revelation of a double-stranded filament exceptionally interesting. However, the cryo-EM images reveal long filaments and the SEC shows a broad peak of sizes. Further, the comparison with the DAX domain and its associated SEC-MALS data (7-8 monomers per polymer) suggest much longer filaments in solution for the DIX domain. Did they do SEC-MALS for the DIX domain? I imagine the answer is more nuanced than is currently implied, and the authors should discuss this more clearly.

Figure 4 and subsection “Axin DAX oligomerizes through a head-to-tail interface but does not form large polymers”. The current work contrasts in some important ways with recent published work from the Bienz lab and colleagues (Yamanishi et al., 2019). Both report affinities for DIX:DIX, DAX:DAX, and DIX:DAX interactions, but the results are different. In one case DIX:DIX affinities are highest and in the other DAX:DAX affinities are highest. This issue comes back again in the discussion of the work in Figure 5, which is interpreted to suggest that DAX:DIX interactions are weaker than either homomeric interaction, something not supported by the Yamanishi data. This is quite important for their modeling and while they do mention it, they don't offer any thoughts on this difference. More discussion of the Yamanishi data would inform the work.

Figure 5A,B. The authors state "Surprisingly, addition of Axin DAX reduced Dvl2 DIX filament size to the point that they no longer sediment (Figure 5A,B)." However, the data presented are much more subtle/nuanced. Even at a 5:1 DAX:DIX ratio, 40% of the DIX domain still pellets. These data contrast with their Yes/NO conclusion and with the data from TIRF. Their conclusion that DAX disrupts DIX filaments is solid but not as clear-cut as it is portrayed. This then is codified in speculative Figure 5D, which needs to be more clearly portrayed as one potential model of many, especially as it presents a quite different view than the section head which suggests filament "capping", not insertion into an existing filament.

Subsection “Avidity for Axin enhances signaling”: Figure 4D. The experiments here are quite interesting but I'd suggest pulling the data out of Figure 4D and making a stand-alone Figure with diagrams illustrating the nature of the complex constructs tested here.

Figure 6 and subsection “The size of Dvl oligomers is limited in cells”. Here the authors try to bridge their lovely biochemical and structural data with imaging data in vivo. This is also a place where caveats are important. The authors clearly show that in this system, with Dvl expressed at near endogenous levels, that it does not form "large" cytoplasmic puncta. This is solid, though most current models would predict that biologically relevant Dvl puncta would likely be at the membrane. The TIRF experiments provide a means to search for these. Here there are clearly "small" spots, which the authors maintain contain "small numbers of GFPs". Their use of single molecule tracking to estimate GFP number in particles was interesting, but needs to be better explained for the reader. Further, they are constrained to look at the apical surface and it is unclear, at least from this work, where Wnt receptor complexes are located and thus if these spots reflect "active" Dvl.

Places where models could be tempered or additional information included:

Introduction. The Introduction could be broadened, to more clearly set the stage of the new work. Given the participation of the Bienz lab and the subsequent focus of the paper on multimerization, I was surprised that the Introduction did not more clearly lay out the previous view that the "signalosome" is a multimeric entity, referencing in more detail the previous literature. On a more minor point they should mention that Axin the phosphorylated tail of LRP5/6 also participates in recruiting Axin into the signalosome. The other area that could be included is a contrast with the active destruction complex, highlighting work by the Bienz lab and others suggesting that APC and Dsh may compete for Axin binding, with APC promoting destruction complex assembly and Dsh its disassembly.

The authors in their title and Abstract emphasize the idea of "Limited Dishevelled/Axin oligomerization" and that "Dvl forms oligomers typically <10 molecules at endogenous expression levels." The degree to which the Wnt receptors and the active destruction complex form larger scale entities, perhaps biomolecular condensates, is an area of active discussion in the field, and, in interests of full disclosure, one in which our lab has a direct interest. I think a revised discussion that more directly addresses different views in the field in light of this new data would be appropriate. Some issues to address. 1) In support of their data, previous work by the Wehrli lab provided a surprising and often ignored data point-an Axin mutant lacking its DIX domain was still functional in mediating β-catenin destruction (at least as assessed by cell fates) but could not be turned OFF. This supports the idea that the DIX:DAX interaction is key to turning down the destruction complex. 2) As the authors note in the last paragraph of the Discussion and as the Bienz lab has previously argued, considerable data support the idea that activated Wnt receptors form a "signalosome", perhaps by phase transition. By focusing so much on the limited size of Dvl oligomers, they miss a chance to discuss this work. These data now include imaging of endogenous Dvl puncta in Wnt-ON cells in *Drosophila* embryos, and its co-localization in puncta in Wnt on cells with Axin which it is expressed at near-endogenous levels (3x wildtype; Schaefer et al., 2018). 3) One possibility that would reconcile different views is that oligomerization is both sharply concentration dependent and also regulated in vivo. Their work, while lovely, largely relies on in vitro studies of the isolated DIX and DAX domains, and does not include either other domains of the same protein or their in vivo partners. Work from the Bienz lab and our lab suggest, for example, that Dsh and APC exert contrasting effects on the "stability" of Axin polymers. Data from Schaefer et al., 2018, suggest that when expressed in vivo at 2-3 fold endogenous concentrations, Axin forms complexes containing tens to hundreds of molecules, and that Wnt signaling regulates this assembly state. The authors should be free to state their own views and argue their position, but I think a more complete discussion that lays out these contrasting views would be more informative for the reader.

---

## [Author Response]

Reviewer #1:In this study, Kan et al., characterized the structural properties of the DIX domains from Dvl2 and Axin. Using cryo-EM, the authors determined the oligomer structure of Dvl2 DIX and revealed that Dvl2 DIX form a two-stranded, antiparallel oligomer. The cryo-EM structure also reveals the molecular basis governing the inter-strand interactions that stabilize the double strand conformation of Dvl2 DIX. The authors went on to discover that the Axin DIX domain (DAX) can break/sever Dvl2 DIX oligomer via capping the ends of the Dvl2 DIX oligomer, which is a rather unexpected finding. The authors showed that formation of small Axin/Dvl2 DIX heteroligomer is critical for the transcriptional activity of Dvl2 in a cell-based assay. Finally, the authors provided evidence showing that in living cells the endogenous Dvl2 only form small oligomers (with 10 or less protomers) regardless of Wnt signal stimulation. With these findings, the authors proposed a new model showing how DIX domain-mediated Dvl/Axin oligomerization can transduce Wnt signaling.The results presented in the current study are potentially interesting and valuable for understanding Wnt signaling. However, the manuscript contains a number of issues as detailed below. These issues dampen the rigorous of the conclusions drawn, and thus will need to be addressed.Major issues:1) Figure 1B and C: the pellet fraction of Dvl2 DIX was observed to increase as a function of protein concentration decrease at the protein concentrations below 6.25 µM. Why could this occur? Related to the above point, the dilution experiment shown in Figure 1D is also strange. At the intermediate dilution folds (e.g. around the final concentration ~3.7 µM), one can see obvious depolymerizations of Dvl2 DIX oligomer. Such depolymerization seems to be less obvious at both high and low final concentrations of the dilution experiment.

This has been a concern of ours, and we believe that at low concentrations there is non-specific pelleting of the DIX protein, as we found that addition of 0.05% Tween reduced this background. However, the Tween affected the sedimentation of filaments at some intermediate concentrations (e.g. 25 μM) for reasons that are not clear, as points below and above this were essentially the same as what we observed without Tween. Although going to longer centrifugation times seemed to alleviate this problem, we did not have time to systematically vary different conditions (we are now shut down due to the COVID-19 restrictions). In any case, comparison of the formation assays run with (shown in Author response image 1) and without (Figure 1) Tween show that the Tween does not significantly alter the concentration-dependence of filament formation. Thus, while we are including a plot quantifying the replicates done with Tween in this response (see Author response image 1), we feel confident in leaving the original data from the experiments done in the absence of Tween in the main text. This provides a more direct comparison with the DAX solubilization assays shown in Figure 6, which were also done without detergent. We have also removed the lowest-concentration data point in the formation assay from both the plot and the corresponding gel that was shown in the original submission, as the experiments done in the presence of Tween make it clear that the pelleting at that point is nonspecific.

**Author response image 1. respfig1:** Summary of filament formation data in the presence of 0.05% Tween.

In contrast to the formation assay, the addition of Tween seems to affect the filament dilution assay substantially. We decided it would best to remove the dilution assay from the paper. While we can no longer compare the stability of the filaments to dilution with the concentration at which they form, we now note, based on other data, that large assemblies can form at concentrations only a few-fold above the dimerization KD, suggesting that the lateral contacts are reinforcing the head-to-tail oligomers as they form and allowing them to grow larger than they otherwise would be expected to at those concentrations.

We also note the rather crude nature of these assays, as well as the TIRF assay shown in Figure 6: we do not know what size of filament actually sediments, likewise when the size becomes so small that it gives background rather than distinct puncta in that TIRF assay. This is summarized in point #7 below and is now discussed in subsection “Axin DAX caps Dvl DIX oligomers” in the context of the DAX disruption assay.

2) Figure 2C: The labeling of the figure needs to be improved. It will be helpful for recognizing the head and tail of each subunit in Figure 2C if the secondary structure unit of one protomer (e.g. the brown one at the upper left corner) can be labeled. Similarly, it would help if the strand names containing Y27 and K68 in Figure 2D are labeled. Similarly, secondary structures in Figures 3B and D should be labeled for easy reading.

We thank the reviewer for recommending this clarification. We have amended these figures (note that the old Figure 3D is now 4C).

3) Figure 3 and 4: The SEC analysis in the two figures are complicated by the cleaved MBP in the mixtures. For example, the authors state that the Y27D mutant of Dvl2 DIX is a monomer, but the elution peak overlaps with the MBP tag, which is considerably larger than the monomeric Dvl2 DIX. The authors should remove MBP from the cleaved protein mixtures and re-analyze the elution behaviors of the mutants.

While the relatively large height of the MBP peak (due to its extinction coefficient being 7-8x greater than that of a DIX domain) prevents us from seeing full baseline separation between the MBP peak and some of the DIX domains, we respectfully disagree with the reviewer’s assertion that this complicates interpretation for most of the traces. The SEC traces clearly show that while MBP exhibits a broad elution profile due to its high A280, its profile is still manifestly Gaussian, and its center is at the correct volume (18 mL on our Superose 6 or 11 mL on our Superdex 75 columns). The DIX and DAX peaks are well, if not completely, separated from the MBP peak. The WT DIX filament peak in Figures 1B, 3A and 4A, as well as the DAX NG and DAX NQ/NG peaks in Figure 4A, show complete baseline separation from the MBP peak. The WT DAX peak (Figures 4A and Figure 3—figure supplement 1A) and DIX M60A and N82D peaks (Figure 3A) show nearly complete separation from the MBP peak, if not quite full baseline separation. The DIX Y27D, DIXDC, and DIXY27D+DC peaks in Figure 3—figure supplement 1B show less separation from the MBP peak, but are still clearly distinct, being centered at 13.8 mL, 13.1 mL, and 13.8 mL, respectively – a substantial shift from the MBP peak centered at 11 mL. Furthermore, the gels in Figure 3—figure supplement 1C also show clear separation between MBP and those three proteins, as well as between DIXDE and MBP (which is less clear in the chromatogram).

Nonetheless, we agree that the interpretation of the G65D DIX and DIXDE mutants was difficult given their strong overlap with the MBP peak, though the gel for the DIXDE peak does show substantial separation. We have therefore re-run these proteins, as well as the Y27D mutant and WT DAX, on SEC after depletion of contaminating MBP. As we show in Figure 3—figure supplement 1A, these proteins separate from MBP and run at the same volume on Superdex 75 as before regardless of the amount of contaminating MBP.

4) Figure 3: The G65D mutant is interesting as the mutation seems to disrupt both inter- and intra-strand protomer DIX interaction. This brings up two issues: the mutation cannot really be used to support the inter-strand interaction observed in the cryo-EM structure; and the mutation might have a large overall folding perturbation of the Dvl2 DIX protomer. The authors should check the folding of the G65D mutant. Additionally, the G65D mutant, though not capable of forming DIX oligomers, showed a comparable level of transcriptional activity to that of the M60A mutant (Figure 3C). This will need to be explained.

We are puzzled by the reviewer’s assertion that the G65D mutation completely disrupts oligomer formation. The fact that G65D DIX appears to run larger than MBP on the Superose 6 (Figure 3A) when injected at 180 μM, whereas monomeric mutants such as Y27D clearly run smaller, strongly suggests that its oligomerization is not completely ablated. In the new experiments, we ran partially purified G65D at 60 μM on the Superdex 75 (Figure 3—figure supplement 1A), and it eluted later than MBP but earlier than DIXDE, DIXDC or the Y27D mutant (Figure 3—figure supplement 1) and possessed the characteristic tail reflecting exchange among multiple oligomeric species, suggesting that it can still oligomerize. Furthermore, the fact that the G65D mutant signals substantially better than the known monomer mutant Y27D (Figure 3C) or the F43S mutation that likely unfolds the protein (see Figure 1C in Schwarz-Romond et al., 2007) indicates that it is neither monomeric nor unfolded. The fact that it runs as a moderately sized peak on gel filtration and not as an aggregate further argues against the G65D mutant being unfolded. Rather, the mutation more likely acts by forcing the repacking of the β3-β4 loop due to the native G65 occupying a Ramachandran space that would not be allowed for an aspartate.

Subsection “DIX forms an antiparallel double helix that stabilizes filaments”, third paragraph we note that we cannot rule out that there is some coupling between mutations in the inter-strand interface and the head-to-tail intra-strand interface, which might explain differences in the apparent sizes of the oligomers formed by these mutants.

5) Figure 4 and paragraph describing data in Figure 4 (subsection “Axin DAX oligomerizes through a head-to-tail interface but does not form large polymers”) is confusing. If the pink elution profile was obtained for DAX at 180 µM, why the highest concentration tested was only 24 µM? Even for the lowest concentration (0.7 µM) analyzed, the peak would likely be an ensemble average of different oligomerization states of DAX instead of simple monomer/dimer equilibrium. If this is true, estimation of apparent Kd of 0.7 μm would not be appropriate. Consequently, a number of places in the manuscript concerning the affinity of DAX association will need to be re-written.

We apologize for this confusion. The two experiments refer to different concentration measurements. The 180 μM concentration refers to the initial loading concentration of the protein prior to injection onto the gel filtration; the subsequent run would dilute the protein roughly twelvefold (i.e., to approximately 15 μM). The 24 μM curve was indeed the highest concentration test for the separate SEC-MALS experiment and refers to the final concentration of the eluted protein in the peak as measured by its refractive index (and confirmed by A280). We have now clarified this in subsection “Axin DAX oligomerizes through a head-to-tail interface but does not form large polymers”.

Regarding the rough KD estimate, we agree that assuming that an absolute mass corresponding to 1.5 protomers/oligomer implied a 50:50 mix of monomer and dimer was inappropriate and thank the reviewer for pointing this out. We recalculated the KD including higher-order species and in so doing also discovered another error in our approximation; namely, we had assumed 710 nM to be the concentration of both monomer and dimer, whereas it is actually the total concentration of DAX protomers. A table summarizing the calculated Kd with higher-order species added is shown with the full calculation in Table 3. This shows that adding in higher-order species would not bring the Kd to even 1 μM, thus confirming that DAX-DAX is stronger than DIX-DIX or DIX-DAX, a key conclusion of the biochemical work.

6) Figure 3—figure supplement 1C and D (related to the question concerning Figures 3 and 4): It appears that each cleaved Dvl2 DIX mutant was eluted as an oligomer and a smaller multimer on SEC column. This is a bit odd. Do these two populations exchange? Or each population is trapped in their kinetically stable form?

We apologize if the reviewer was confused by our figures. As shown on the Figure 3—figure supplement 1C gels, the earlier peak (before the MBP) corresponds to uncleaved protein remaining after incomplete TEV digestion. These peaks also disappear when samples are run after depletion of MBP and uncleaved protein (Figure 3—figure supplement 1A).

7) Figure 5A: why ~40% of Dvl2 DIX was recovered in the pellet/filament form even in the presence of very high concentration of DAX? Related to this question, it would be informative to perform an SEC analysis of Dvl2 DIX with increasing concentrations of DAX (or Y760D DAX) to check the hetero-DIX oligomer formation status.

We have added an explanation: “The fact that approximately 40% of the DIX still sediments even with a fivefold molar excess of DAX is likely due in large part to the relatively weak DIX-DAX KD of 10-20 μM (Yamanishi et al., 2019a). Furthermore, DAX insertions near the ends of DIX filaments would consume DAX while only shifting a small portion of the DIX contained in that filament from the pellet to the supernatant. Likewise, DAX insertion into a DIX oligomer that is already too small to sediment (the likelihood of which would increase as the number of severed DIX oligomers increased) would consume DAX without shifting any DIX from the pellet to the soluble fraction. “

The SEC experiment would be an excellent way to confirm the DAX-mediated solubilization of DIX filaments observed by sedimentation and TIRF. However, the 10-15x on-column dilution inherent in an SEC experiment requires extremely high input concentrations of each partner, which makes the experiment difficult. However, we note that the TIRF assay provides an orthogonal readout of the same phenomenon. Instead of SEC, we have added a new experiment using native gel shifts of labeled filaments (subsection “Axin DAX caps Dvl DIX oligomers” and Figure 5—figure supplement 2), which shows that at the same concentrations used for the sedimentation assay shown in Figure 5, a shift of the DIX oligomer population to a smaller, faster migrating species is apparent as DAX is added.

8) The data for the "Avidity for Axin enhances signaling" and Figure 4D is not convincing and interpretation is tentative. The data seem to indicate that as long as Dvl2 and Axin can form heteromer, the Dvl2 activity can be realized. Additionally, the enhanced heteromer formation between Dvl2 and Axin seems to promote Dvl2 transcription activity. The use of TNKS2-SAM potentially brings in another indirect binder to the system. The interpretation presented in the current manuscript based on the data at the present form is one of many possibilities. It would be nice to use a pair of small heterodimer forming protein domains to replace Dvl2 and Axin DIX domains, respectively, and then to repeat the transcription assay. This is a demanding experiment, the authors should at least consider similar line of assay or any assay that would provide a more definitive result to support their model.

This is now Figure 6. We disagree with the reviewer’s assertion that formation of an Axin:Dvl2 heterodimer alone is sufficient for Wnt signaling. The Dvl2 Y27D can still bind Axin 1:1 (we now reference the Yamanishi et al., 2019 paper on this point; subsection “Avidity enables Axin recruitment by Dvl2 to drive signaling”) but is completely inactive in Wnt signaling. Furthermore, the Sm1-DIX Y27D Dvl2 should heptamerize and could bind up to one Axin per Dvl (i.e., 7 Axin per heptamer), but it is still completely inactive as well. Thus, an enhanced avidity for Axin mediated by oligomerization of the Dvl2 DIX domain appears to be absolutely essential for Wnt signal transduction.

We agree that being able to perturb both Axin and Dishevelled simultaneously would be ideal. However, we do not have a quintuple knockout cell line lacking all three Dvl isoforms and the two Axin isoforms that would allow us to express Axin variants as well as Dvl variants at nearendogenous levels without contaminating endogenous protein, as we have done here for Dvl2 using the TKO cells. Results with overexpressed proteins and the presence of the endogenous versions of the proteins would not be readily interpretable. We do plan to develop and use such cells in the future, but this is beyond the scope of this work.

Finally, we have rewritten the first paragraph of subsection “Avidity enables Axin recruitment by Dvl2 to drive signaling” to further clarify these points.

Reviewer #2:Wnt signaling is one of the handful of signaling pathways that play key roles in embryonic development, tissue homeostasis and which go wrong in cancer. While the field created an outline of the signaling pathway two decades ago, many unanswered questions remain. The key regulated effector is the protein β-catenin, which, in the absence of Wnt ligands, is targeted for phosphorylation and ultimate proteolytic destruction by the multiprotein destruction complex. One key open question involves the molecular mechanisms by which Wnt ligands turn down destruction complex activity. This requires the cytosolic effector Dvl, but its mechanism of action remains unclear. One critical issue is whether and how Dsh:Dsh, Axin:Axin, and Dsh:Axin multimerization regulate the balance of destruction complex activity. Here Weis and colleagues present the results of a multidisciplinary tour-de-force investigation, combining cryo-EM, powerful analytical biochemistry, and in vivo functional and imaging observations. They present some very intriguing new insights, revealing the polymerization of Dsh's DIX domain into anti-parallel double helical filaments, careful investigation of the parameters of polymerization in the presence and absence of Axin, clever structure-based mutational analyses that reveal important features of homo- and hetero-polymerization, combined with in vivo assays of function. These data provide a firm biochemical basis and important molecular constraints for previous speculative models for Dsh/Axin interactions, significantly advancing our understanding of this key signaling pathway. In general, their data are lovely, well controlled and quantitatively analyzed, and their conclusions well founded. As I lay out below, I think at times they are a bit to enamored with their in vitro data, making broad conclusions about events in vivo that could be presented in a more nuanced way by considering other data in the field. With these and some other minor modifications, I think this work will be of exceptional interest to cell, developmental and cancer biologists.Data issuesFigure 1. This was lovely work but I found one thing puzzling. Why was the pelleting profile of Dsh biphasic, with significant pelleting at low concentrations (<1µm)?

Please see the response to reviewer 1’s point 1.

Figures 2-4. As I will note below in comments on the model, the authors come down very firmly on the idea that the "active" Dsh is in short polymers of less than 10 monomers. Their structural studies are stunning, and the revelation of a double-stranded filament exceptionally interesting. However, the cryo-EM images reveal long filaments and the SEC shows a broad peak of sizes. Further, the comparison with the DAX domain and its associated SEC-MALS data (7-8 monomers per polymer) suggest much longer filaments in solution for the DIX domain. Did they do SEC-MALS for the DIX domain? I imagine the answer is more nuanced than is currently implied, and the authors should discuss this more clearly.

We do not mean to give the impression that the ability of the DIX domain in isolation to form large polymers reflects the situation in vivo. A strength of the cellular counting presented here is the careful attention to maintaining labeled Dvl2 at near-endogenous levels. The formation of filaments in vitrodoes, however, reflect fundamental properties of DIX-DIX interactions: previously reported mutational data, as well as those presented here, make clear that the intraand inter-strand interactions observed in the structure are relevant. However, the size of oligomers *in cellulo* is considerably smaller than those observed with the purified DIX domains. This could reflect several things. The concentrations of Dvl and Axin estimated in cells (~150 nM) is much lower than those used in our biochemical experiments. Second, we discuss that other parts of Dvl may well contribute to limiting the size of oligomers in vivo.

We have examined the DIX filaments by SEC-MALS. As now shown in Figure 1B, the filaments elute in a broad distribution ranging from 4.9-9.5 MDa. Unfortunately, for reasons of detection this experiment cannot be done at near cellular concentrations.

Figure 4 and subsection “Axin DAX oligomerizes through a head-to-tail interface but does not form large polymers”. The current work contrasts in some important ways with recent published work from the Bienz lab and colleagues (Yamanishi et al., 2019). Both report affinities for DIX:DIX, DAX:DAX, and DIX:DAX interactions, but the results are different. In one case DIX:DIX affinities are highest and in the other DAX:DAX affinities are highest. This issue comes back again in the discussion of the work in Figure 5, which is interpreted to suggest that DAX:DIX interactions are weaker than either homomeric interaction, something not supported by the Yamanishi data. This is quite important for their modeling and while they do mention it, they don't offer any thoughts on this difference. More discussion of the Yamanishi data would inform the work.

This speaks to a key result of our work, that the DAX:DAX affinity is stronger than either DIX:DIX or DIX:DAX. Per reviewer 1’s point #5, we have more rigorously estimated the KD for DAX:DAX (See Table 3), and it is sub-μM. Regarding the Yamanishi et al. paper, we suspect that the M3 (I758A/R761D) DAX mutant that was used in place of the more typical M4 head mutant (Y760D) may have affected the measurements, which were made by fluorescence anisotropy. Specifically, they observed a substantial increase in anisotropy even for the two mutants that should not bind each other, and there was also a stark difference in the anisotropy changes for DIX:DIX (~700) and DAX:DAX (~800) as compared to DAX:DIX (~200) and DIX:DAX (~350). If the domains are all roughly the same size and shape, the anisotropy change would be expected to be roughly equal. Our data were derived from unmutated and unmodified protein, so we believe that our values are more reliable.

Figure 5A, B. The authors state "Surprisingly, addition of Axin DAX reduced Dvl2 DIX filament size to the point that they no longer sediment (Figure 5A,B)." However, the data presented are much more subtle/nuanced. Even at a 5:1 DAX:DIX ratio, 40% of the DIX domain still pellets. These data contrast with their Yes/NO conclusion and with the data from TIRF. Their conclusion that DAX disrupts DIX filaments is solid but not as clear-cut as it is portrayed. This then is codified in speculative Figure 5D, which needs to be more clearly portrayed as one potential model of many, especially as it presents a quite different view than the section head which suggests filament "capping", not insertion into an existing filament.

Please see the response to reviewer 1, point 7. In the in vitroexperiments in Figure 5 and Figure 5—figure supplements 1, 2 and 3, we start with preformed filaments and observe reduction in their size, implying severing is occurring. Figure 5D is meant to describe a model for what we see in vitroand reflects one of the possible models for what happens in vivo, but certainly not the only one. We do not mean to be dogmatic about whether, in cells, DAX caps DIX filaments or inserts into and severs existing ones. The capping model is perhaps simpler, but one possible interpretation of the effect of the groove mutants (Figure 5—figure supplement 3) is hyperstabilization, which would require insertion to be important (albeit potentially in addition to capping and not in place of it). While we never observed larger assemblies in the cell imaging, we cannot rule out the possibility that they formed transiently on a timescale faster than that of our image acquisition. We have tried to clarify these points in the Discussion (paragraph 3).

Subsection “Avidity for Axin enhances signaling”: Figure 4D. The experiments here are quite interesting but I'd suggest pulling the data out of Figure 4D and making a stand-alone Figure with diagrams illustrating the nature of the complex constructs tested here.

We agree, and it is now a standalone figure (Figure 6).

Figure 6 and subsection “The size of Dvl oligomers is limited in cells”. Here the authors try to bridge their lovely biochemical and structural data with imaging data in vivo. This is also a place where caveats are important. The authors clearly show that in this system, with Dvl expressed at near endogenous levels, that it does not form "large" cytoplasmic puncta. This is solid, though most current models would predict that biologically relevant Dvl puncta would likely be at the membrane. The TIRF experiments provide a means to search for these. Here there are clearly "small" spots, which the authors maintain contain "small numbers of GFPs". Their use of single molecule tracking to estimate GFP number in particles was interesting, but needs to be better explained for the reader. Further, they are constrained to look at the apical surface and it is unclear, at least from this work, where Wnt receptor complexes are located and thus if these spots reflect "active" Dvl.

In our epifluorescent imaging (see supplementary information), which was done in addition to TIRF imaging, our depth of focus is on the order of the height of the cells – thus we mostly do not distinguish cytoplasmic from membrane associated puncta in that measurement and we now simply note that we do not observe the multiple large puncta reported in overexpression studies (subsection “The size of Dvl oligomers is limited in cells”). We have also expanded the description of the image analysis in the Materials and methods section.

Places where models could be tempered or additional information included:Introduction. The Introduction could be broadened, to more clearly set the stage of the new work. Given the participation of the Bienz lab and the subsequent focus of the paper on multimerization, I was surprised that the Introduction did not more clearly lay out the previous view that the "signalosome" is a multimeric entity, referencing in more detail the previous literature. On a more minor point they should mention that Axin the phosphorylated tail of LRP5/6 also participates in recruiting Axin into the signalosome. The other area that could be included is a contrast with the active destruction complex, highlighting work by the Bienz lab and others suggesting that APC and Dsh may compete for Axin binding, with APC promoting destruction complex assembly and Dsh its disassembly.

In the Introduction (paragraph two) we have added a more precise definition of the signalosome as a crosslinked complex involving the receptors and their cytoplasmic effectors. Regarding the proposal that phosphorylated LRP5/6 tails can bind to Axin, we know of no direct rigorous demonstration that phosphorylated LRP5/6 tails can bind to Axin. The possible competition with APC was also noted at the end of the following paragraph.

The authors in their title and Abstract emphasize the idea of "Limited Dishevelled/Axin oligomerization" and that "Dvl forms oligomers typically <10 molecules at endogenous expression levels." The degree to which the Wnt receptors and the active destruction complex form larger scale entities, perhaps biomolecular condensates, is an area of active discussion in the field, and, in interests of full disclosure, one in which our lab has a direct interest. I think a revised discussion that more directly addresses different views in the field in light of this new data would be appropriate. Some issues to address. 1) In support of their data, previous work by the Wehrli lab provided a surprising and often ignored data point-an Axin mutant lacking its DIX domain was still functional in mediating β-catenin destruction (at least as assessed by cell fates) but could not be turned OFF. This supports the idea that the DIX:DAX interaction is key to turning down the destruction complex. 2) As the authors note in the last paragraph of the Discussion and as the Bienz lab has previously argued, considerable data support the idea that activated Wnt receptors form a "signalosome", perhaps by phase transition. By focusing so much on the limited size of Dvl oligomers, they miss a chance to discuss this work. These data now include imaging of endogenous Dvl puncta in Wnt-ON cells in *Drosophila* embryos, and its co-localization in puncta in Wnt on cells with Axin which it is expressed at near-endogenous levels (3x wildtype; Schaefer et al., 2018). 3) One possibility that would reconcile different views is that oligomerization is both sharply concentration dependent and also regulated in vivo. Their work, while lovely, largely relies on in vitro studies of the isolated DIX and DAX domains, and does not include either other domains of the same protein or their in vivo partners. Work from the Bienz lab and our lab suggest, for example, that Dsh and APC exert contrasting effects on the "stability" of Axin polymers. Data from Schaefer et al., 2018, suggest that when expressed in vivo at 2-3 fold endogenous concentrations, Axin forms complexes containing tens to hundreds of molecules, and that Wnt signaling regulates this assembly state. The authors should be free to state their own views and argue their position, but I think a more complete discussion that lays out these contrasting views would be more informative for the reader.

We have expanded the Discussion to include these points, some of which were in the original version but not elaborated upon.